# History-Bootstrapped Flow Matching for Inverse Boiling Reconstruction

**Xianwei Zou** [1]  **Sheikh Md Shakeel Hassan** [1]  **Arthur Feeney** [1]  **Aparna Chandramowlishwaran** [1]

## Abstract

Reconstructing spatiotemporal fields from partial observations is fundamental to scientific inference, from inferring atmospheric states from satellite data to recovering fluid states from imaging. When observations are incomplete, the inverse problem is fundamentally ill-posed: even when the underlying PDE dynamics are Markovian in the full state, partial observation operators induce a non-Markovian posterior that cannot be resolved from a single timestep. We propose a history-bootstrapped autoregressive flow matching (HB-ARFM) for spatiotemporal inverse reconstruction under partial observability. Observation history bootstraps the initial reconstruction via conditional flow matching, reducing ambiguities. The same conditional transport model is then applied autoregressively, conditioning on both new observations and past predictions to propagate the reconstruction forward in time. We evaluate the method on boiling dynamics reconstruction, recovering full velocity and temperature fields from interface geometry and motion. Across two inverse tasks with varying observation sparsity, HB-ARFM produces physically and temporally valid reconstructions where other models fail.

## 1. Introduction

Two-phase boiling is the most efficient known form of heat transfer. It plays a central role in energy systems and thermal management, with immense real-world applications ranging from data center cooling (Azarifar et al., 2024) and power generation (Dirker et al., 2019) to renewable energy (Ni et al., 2024) and space thermal systems (Sielaff et al., 2022). Few challenges in science have such broad industrial and societal consequences, yet boiling remains notoriously

difficult to characterize experimentally. Many of the quantities that govern the performance of two-phase flows, such as temperature and velocity, mass transfer between phases, and interfacial heat flux are either unobservable or extremely challenging to measure in situ, particularly near rapidly evolving liquid-vapor interfaces. As a result, much of our understanding of boiling relies on numerical simulation (Dhir et al., 2013) and correlation studies (Bertsch et al., 2009), even though high-speed video imaging data have existed for decades (Gaertner, 1965; Seong et al., 2023; Suh et al., 2024). Prior learning-based approaches to boiling and multiphase flow focus primarily on forward prediction or surrogate modeling from fully observed simulations, including neural operators and supervised regression frameworks (Hassan et al., 2023; 2025; Khodakarami et al., 2025b).

This gap gives rise to a canonical inverse problem: reconstructing the full thermofluid state of a boiling flow from sparse, image-derived observations of the interface geometry and motion. While the governing equations of two-phase flow are Markovian in the full state (Prosperetti & Tryggvason, 2009), the inverse problem induced by experimental observability is not. Partial observations do not uniquely determine the hidden velocity and temperature fields, and single-timestep measurements are insufficient to enforce the temporal and physical constraints required for consistent reconstruction. To date, the inverse reconstruction of full spatiotemporal boiling fields from imaging alone has remained largely unexplored.

Recent advances in generative modeling, particularly diffusion models (Ho et al., 2020; Song et al., 2021) and flow matching (Lipman et al., 2023), have demonstrated remarkable success in high-dimensional inverse PDE problems (Holzschuh et al., 2023; Huang et al., 2024; Jacobsen et al., 2025; Yao et al., 2025). However, these methods typically address snapshot or static reconstruction, producing reconstructions that violate temporal consistency. Extensions to video and spatiotemporal data have focused primarily on joint spatiotemporal inverse modeling (Li et al., 2024) or forward prediction: generating future frames given complete past observations (Jin et al., 2025; Gao et al., 2025). When adapted to inverse problems, such autoregressive models encounter a cold start problem without ground truth initial conditions. These limitations are not specific to boiling, but boiling provides an especially stringent test due to sharp

[1]Department of Electrical Engineering and Computer Science, University of California Irvine, USA. Correspondence to: Xianwei Zou <xianwz2@uci.edu>, Aparna Chandramowlishwaran <amowli@uci.edu>.

*Proceedings of the 43rd International Conference on Machine Learning*, Seoul, South Korea. PMLR 306, 2026. Copyright 2026 by the author(s).

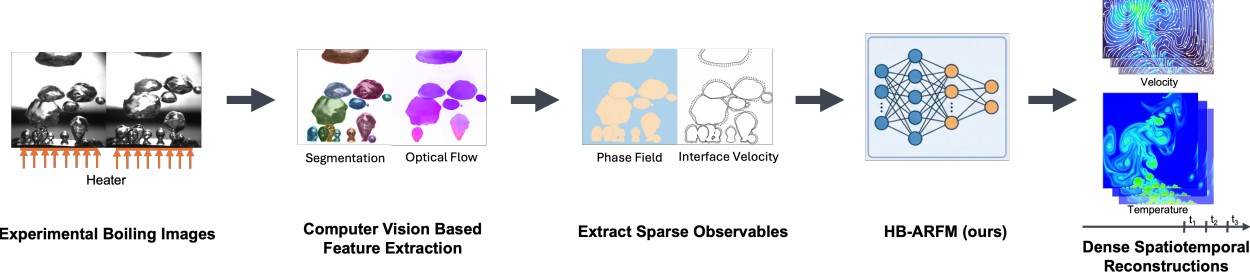

*Figure 1.* **An Example Pipeline of Spatiotemporal Boiling Dynamics Reconstruction from High-speed Imaging.** Sequential high-speed images are processed through segmentation and optical flow to extract phase field and interface velocity. These extracted observations condition our proposed HB-ARFM model which then outputs full spatiotemporal predictions of velocity and temperature fields, enabling complete multiphase fluid dynamics reconstruction from raw imaging data alone.

interfaces, multi-physics coupling, and strong sensitivity to hidden transport processes.

In this work, we address this challenge by using temporal history to bootstrap the initial state and reduce inverse ambiguity. We make the following core contributions:

1. **History-bootstrapped autoregressive flow matching for spatiotemporal inverse problems.** We propose a unified inverse reconstruction model that reduces the intrinsic non-Markovianity induced by partial observability by using observation history to initialize latent states, followed by conditional autoregressive propagation within a single flow matching formulation.

2. **First reconstruction of full boiling thermofluid fields from imaging-derived observations.** We demonstrate, for the first time, the spatiotemporal reconstruction of complete velocity and temperature fields in multiphase boiling flows using only interface geometry and motion (see Figure 1), substantially improving physical consistency and heat flux estimation under partial observation.

3. **Boiling as a stress test for inverse SciML models and identification of failure modes.** Through two inverse reconstruction tasks of varying difficulty, we systematically expose failure modes of existing generative, autoregressive, and physics-informed models in the presence of moving interfaces, multi-physics coupling, and partial observability, establishing boiling as a challenging benchmark for inverse modeling in scientific machine learning.

## 2. Problem Formulation

We consider a spatiotemporal system governed by PDEs:

$$\partial_t X(t, x) = \mathcal{F}(X(t, \cdot))(x), \quad x \in \Omega \subset \mathbb{R}^d, \quad (1)$$

where $X(t, \cdot) : \Omega \to \mathbb{R}^p$ denotes the full physical state of the system at a timestep $t \in [0, T]$. We observe the system through a partial and potentially noisy measurement operator $\mathcal{H}$:

$$y(t) = \mathcal{H}(X(t, \cdot)) + \epsilon(t). \quad (2)$$

**Inverse reconstruction under partial observability.** Given observations $\{y(t)\}_{t=0}^T$, our objective is to reconstruct the spatiotemporal dynamics $\{X(t)\}_{t=0}^T$. This inverse problem is ill-posed: when $\mathcal{H}$ is non-invertible, multiple states $X(t)$ are consistent with the same observation $y(t)$, particularly for quantities absent from measurements. In multiphase fluid dynamics, interface geometry alone does not uniquely determine bulk velocity and temperature fields.

An asymmetry arises from partial observability, with theoretical grounding in the Mori-Zwanzig (MZ) formalism (Mori, 1965; Zwanzig, 1973). When the full state $X(t)$ is projected onto the observable subspace $y(t) = \mathcal{H}(X(t))$, MZ theory establishes that the effective dynamics of $y(t)$ take the form of a generalized Langevin equation comprising a Markovian term, a non-local memory kernel, and an orthogonal fluctuation term. The memory kernel is non-zero whenever hidden variables exist (i.e., whenever H is non-invertible) making the inverse posterior non-Markovian. Instantaneous observation lacks sufficient information to recover the influence of the unobserved degrees of freedom, making reconstruction fundamentally ambiguous.

Standard autoregressive models that condition on previous states $X(t-1)$ assume access to the full Markovian state (Li et al., 2022), which is precisely what we seek to reconstruct. Conversely, methods that reconstruct each frame independently from $y(t)$ ignore the temporal structure induced by the governing dynamics (Huang et al., 2024), and in the MZ sense, discard the memory kernel entirely.

**Challenge.** We seek reconstructions $\hat{X}$ that are (1) observationally consistent: $\mathcal{H}(\hat{X}(t)) \approx y(t)$, (2) physically plausible: preserving structure from the governing equations, (3) temporally coherent: exhibiting smooth and stable evolution, and (4) long rollout: maintaining robust reconstruction over time. The difficulty lies in balancing these objectives: at initialization, no prior state estimate exists to propagate; during temporal evolution, we must avoid both observation drift and error accumulation.

# 3. Method: History-Bootstrapped ARFM

We study the inverse reconstruction of latent physical states $X_t \in \mathbb{R}^d$ from partial observations $y_t \in \mathbb{R}^m$. At a single timestep, the mapping $y_t \mapsto X_t$ is non-invertible since multiple physical states may have the same partial observables. A key observation is that temporal histories reduce this ambiguity. While a single observation is underdetermined, the history $y_{0:t-1}$ constrains the admissible states through implicit physical dynamics. However, conditioning on a full history at every timestep is inefficient and unnecessary once a coherent latent state estimate has been formed.

This motivates a two-stage approach, as shown in Figure 2: (i) a history-conditioned bootstrap reconstruction to infer the first hidden state using observations only, followed by (ii) autoregressive flow matching to propagate the solution forward using the model's own predictions while assimilating new observations. Our goal is unified modeling, so both stages are implemented using a shared conditional flow matching model with time-varying context. Appendix B summarizes the notation used in this section.

## 3.1. History-Conditioned Bootstrap Reconstruction

We assume observations are available starting at $t = 0$. Given a history window of length $w$, the goal is to reconstruct the first latent state at time $t = w$. A temporal encoder $\zeta_\phi$ aggregates the observation history into a bootstrap state: $\hat{\mathbf{X}}_w = \zeta_\phi(y_{0:w})$. Conditioned on $\mathbf{c}_w = [y_w, \hat{X}_w]$, a flow matching velocity field $v_\theta(\cdot, \mathbf{c}_w, s)$ transports samples from a reference Gaussian distribution to the posterior distribution of $X_w$. This history-conditioned reconstruction provides an initialization for autoregressive rollout.

The fixed history window $w$ is justified by the decay rate of the memory kernel. MZ guarantees memory decays on a characteristic timescale beyond which past observations provide exponentially diminishing information about the current state. In boiling, the relevant timescale is set by the bubble rise time and condensation time, both of which are captured within a fixed window length.

## 3.2. Autoregressive Sequential Reconstruction

For timesteps $t > w$, reconstruction proceeds autoregressively. At each timestep, the model conditions on the current observation and the previous predicted latent state, $\mathbf{c}_t = [y_t, \hat{X}_{t-1}]$. Flow matching is then used to sample $\hat{X}_t$ by solving the ODE defined by the velocity field $v_\theta(\cdot, \mathbf{c}_t, s)$. This hybrid conditioning couples inverse reconstruction from partial observations with dynamical propagation via the model's own state estimates. We follow the typical flow matching procedures and loss (Lipman et al., 2023) to train our model. The full training algorithm is summarized in Algorithm 1.

---

**Algorithm 1** Training: History-Bootstrapped ARFM

**Require:** Dataset $\mathcal{D} = \{(X_{0:T}, y_{0:T})\}$, history length $w$, rollout length $K$
**Require:** Velocity network $v_\theta$, history encoder $\zeta_\phi$
1: **while** not converged **do**
2:    Sample trajectory $(X_{0:T}, y_{0:T}) \sim \mathcal{D}$
3:    Sample start time $t_0 \sim \text{Uniform}(w, T - K)$
4:    *// Bootstrap: estimate initial state from history*
5:    $\hat{X}_{t_0} = \zeta_\phi(y_{t_0-w:t_0-1})$
6:    $\mathcal{L}_{\text{boot}} = \|\hat{X}_{t_0} - X_{t_0}\|^2$
7:    *// Flow matching loss at bootstrap frame*
8:    Sample $\mathbf{x}^0 \sim \mathcal{N}(\mathbf{0}, \mathbf{I})$, $s \sim \mathcal{U}(0, 1)$
9:    $\mathbf{x}^s = (1 - s)\mathbf{x}^0 + sX_{t_0}$
10:   $\mathbf{c}_{t_0} = [y_{t_0}, \hat{X}_{t_0}]$
11:   $\mathcal{L}_{\text{boot}} \mathrel{+}= \|X_{t_0} - \mathbf{x}^0 - v_\theta(\mathbf{x}^s, \mathbf{c}_{t_0}, s)\|^2$
12:   *// Autoregressive rollout*
13:   $X_{\text{ctx}} = \hat{X}_{t_0}$
14:   **for** $k = 1$ to $K - 1$ **do**
15:      Sample $\mathbf{x}^0 \sim \mathcal{N}(\mathbf{0}, \mathbf{I})$, $s \sim \mathcal{U}(0, 1)$
16:      $\mathbf{x}^s = (1 - s)\mathbf{x}^0 + sX_{t_0+k}$
17:      $\mathbf{c}_{t_0+k} = [y_{t_0+k}, X_{\text{ctx}}]$
18:      $\mathcal{L}_{\text{AR}} \mathrel{+}= \|X_{t_0+k} - \mathbf{x}^0 - v_\theta(\mathbf{x}^s, \mathbf{c}_{t_0+k}, s)\|^2$
19:      *// Update context for next step*
20:      $X_{\text{ctx}} = X_{t_0+k}$
21:   **end for**
22:   $\mathcal{L} \leftarrow \mathcal{L}_{\text{boot}} + \mathcal{L}_{\text{AR}}/(K - 1)$
23:   Update $\theta, \phi$ via $\nabla \mathcal{L}$
24: **end while**

---

## 3.3. Inference

At test time (Algorithm 2), we first collect observations for $w$ timesteps before reconstruction begins. The initial state $\hat{X}_w$ is reconstructed using the history-conditioned model. The system is then evolved autoregressively for $t = w + 1, \ldots, T$, conditioning on $(y_t, \hat{X}_{t-1})$ at each timestep. Inference requires solving an ODE using the learned velocity field and is fully deterministic given the sampled initial noise variables.

## 3.4. Architecture Design Choices

**FM backbone.** The velocity field $v_\theta$ is parameterized using a Residual U-Net architecture (Zhang et al., 2020), a modified version of the original U-Net (Ronneberger et al., 2015) with residual blocks. The network takes as input the concatenation of: (1) the state $\mathbf{x}_t \in \mathbb{R}^{C_{out} \times H \times W}$, (2) the current observation $y_t$ and (3) $\hat{X}_{t-1}$ either bootstrapped or from prior reconstruction. The flow time $s \in [0, 1]$ is encoded via sinusoidal embeddings and injected into residual blocks. The ODE $d\mathbf{x}_s/ds = v_\theta(\mathbf{x}_s, \mathbf{c}_t, \hat{y}_{t-1}, s)$ is solved using 4th order Runge Kutta.

**History encoder.** The history encoder $\zeta_\phi$ is implemented

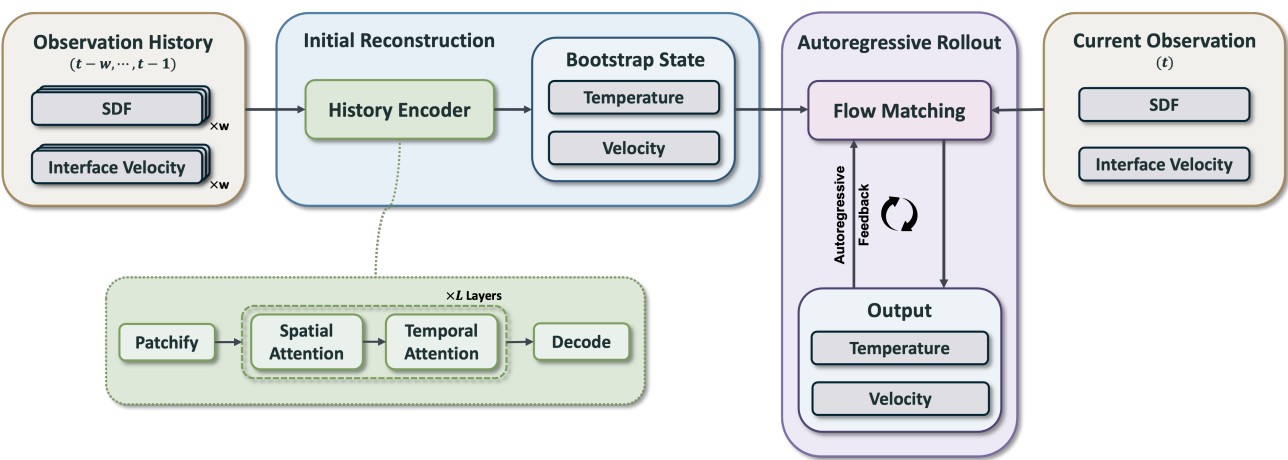

*Figure 2.* **History-Bootstrapped ARFM**. The model combines a history encoder processing temporal sequences with an FM UNet for both initial and AR reconstructions. The FM UNet predicts the full spatiotemporal fields. The green path shows initial reconstruction while the red feedback loop enables sequential reconstruction with data assimilation.

---

**Algorithm 2** Inference: History-Bootstrapped ARFM

---

**Require:** Observations $y_{0:T}$, history length $w$, velocity field $v_\theta$, history encoder $\zeta_\phi$
1: Encode history $\mathbf{c}_w = \zeta_\phi(y_{0:w})$
2: Sample $\mathbf{x}^0 \sim \mathcal{N}(0, I)$
3: Solve ODE to obtain $\hat{X}_w$
4: **for** $t = w + 1$ to $T$ **do**
5:     $\mathbf{c}_t = [y_t, \hat{X}_{t-1}]$
6:     Sample $\mathbf{x}^0 \sim \mathcal{N}(0, I)$
7:     Solve ODE with $v_\theta(\cdot, \mathbf{c}_t, \cdot)$ to obtain $\hat{X}_t$
8: **end for**
9: **Output:** $\{\hat{X}_t\}_{t=w}^T$

---

as a factored space-time transformer that encodes observation history $\{y_{t-w}, \ldots, y_{t-1}\}$ into an initial bulk state estimate. Each conditioning frame is independently patchified via a strided $p \times p$ convolutional embedding, producing a sequence of $N = (H/p)(W/p)$ patch tokens per frame with embedding dimension $d$. Fixed 2D sinusoidal positional embeddings are added spatially, and learned temporal positional embeddings are added across the history axis, giving every token a unique space-time identity. Tokens are processed through $L$ alternating transformer blocks: spatial self-attention operates within each frame (attending over the $N$ patch tokens), and causal temporal self-attention operates across frames for each patch location independently (attending over the $w$ history steps with a causal mask to respect temporal ordering). Each block uses pre-norm, multi-head scaled dot-product attention with FlashAttention, and a SwiGLU feed-forward network. After the final layer norm, only the last frame's tokens are retained and decoded back to pixel space via a linear unpatchify head, followed by learnable scale and bias parameters for training stability.

This factored design allows each spatial location to integrate context from the entire observation history while avoiding the quadratic cost of joint space-time attention.

## 4. Application: Boiling Flow Reconstruction

### 4.1. Physical Setting and Observability

We evaluate our method on the challenging ill-posed inverse problem arising in two-phase boiling flows, in which latent thermofluid states must be inferred from sparse, image-derived observations of an evolving liquid-vapor interface. The interface $\Gamma(t)$ is represented by the zero level-set of a signed distance function $\phi : \Omega \times [0, \infty) \rightarrow \mathbb{R}$, such that $\Gamma(t) = \{\mathbf{x} \in \Omega : \phi(\mathbf{x}, t) = 0\}$, $\phi(\mathbf{x}, t) < 0$ for $\mathbf{x} \in \Omega_\ell(t)$ and $\phi(\mathbf{x}, t) > 0$ for $\mathbf{x} \in \Omega_v(t)$, where $\Omega_\ell$ and $\Omega_v$ respectively denote the liquid and vapor phases. The signed distance function $\phi(\cdot, t)$ satisfies the Eikonal equation $||\nabla\phi||_2 = 1$ over $\Omega$.

The thermofluidic state of the system is given by $\mathbf{X}(\mathbf{x}, t) := (\tau(\mathbf{x}, t), \mathbf{u}(\mathbf{x}, t))$, where $\tau$ and $\mathbf{u}$ are the temperature and velocity defined in $\Omega_\ell \cup \Omega_v$. It satisfies a two-phase system described by incompressible Navier-Stokes equations with phase-dependent properties, an advection-diffusion equation for temperature with latent heat source terms, and interfacial jump conditions enforcing mass, momentum, and energy balance at $\Gamma$. Appendix G details the equations.

In experimental settings, the full state $\mathbf{X}$ is not observable. Infrared thermometry can only measure surface temperatures (Bucci et al., 2016), while particle-based velocimetry techniques (PIV/PTV) face fundamental challenges near vapor-liquid interfaces (Phillips, 2014). Instead, we assume access only to image-derived observations $\mathbf{y}$: (i) the bubble interface geometries encoded by $\phi$, and (ii) the interface

normal velocity $\mathbf{u}_\Gamma$. The former is obtained from segmentation (Hessenkemper et al., 2022; Zhang et al., 2026) and the latter from image sequences via optical flow (Hassan et al., 2023). No direct measurements of the temperature $\tau$ or velocity $\mathbf{u}$ fields are assumed to be available, particularly near the interface where standard diagnostics are unreliable.

Boiling introduces: (i) a sharp interface with topology changes, (ii) tightly coupled scalar temperature and vector velocity fields, (iii) stiff gradients near solid boundaries, and (iv) multi-scale dynamics driven by buoyancy, nucleation, and surface tension. These properties make boiling a novel benchmark for the inverse field reconstruction problem.

### 4.2. Dataset and Inverse Reconstruction Tasks

**Dataset.** We evaluate our method on the BubbleML dataset (Hassan et al., 2023; 2025) for inverse reconstruction. We consider both subcooled pool and flow boiling, which are of significant practical interest due to their applications in immersion and cold plate cooling of electronic systems (Azarifar et al., 2024). See Appendix D for detailed setup.

In subcooled boiling, the bulk liquid is maintained at a temperature below its saturation point, while heat is supplied from below through a constant-temperature heater. As vapor bubbles form and rise from the heated surface, they enter the subcooled bulk liquid and undergo condensation. This process generates high-frequency condensation-induced turbulent vortices, leading to highly transient, multiscale flow dynamics. Predicting these rapidly evolving vortices poses a substantial challenge for machine learning models, particularly in inverse reconstruction settings.

**Inverse Tasks.** We define two inverse sub-problems of different difficulty, varying observation sparsity, and temporal availability to stress-test non-Markovian effects.

1. **Transport reconstruction.** Given the interface geometry $\phi$, reconstruct the temperature field $\tau$. This task requires inferring a hidden transport field from geometric information alone, without any direct thermal measurements (Fu et al., 2026; Khodakarami et al., 2025a).

2. **Coupled flow-transport reconstruction.** Given $\phi$ and interface velocity $\mathbf{u}_\Gamma$, jointly reconstruct $(\tau, \mathbf{u})$ in both phases, coupling momentum and heat transport induced by phase change. This setting also corresponds to sparse flow reconstruction, where only partial kinematic information is observed (Callaham et al., 2019; Sun & Wang, 2020).

### 4.3. Experimental Setup

**Baselines.** We compare **HB-ARFM** against several baseline diffusion models and PDE surrogates including **DDPM** (Ho et al., 2020), **score-based diffusion** (VE-SDE) (Song et al., 2021), **flow matching** (Lipman et al., 2023) conditioned only on $\mathbf{y}_t$ without temporal context, **Diffusion-PDE** (Huang et al., 2024), **PDEDiff** (Shysheya et al., 2024) and PDE-surrogates **Bubbleformer** (Hassan et al., 2025), a transformer model for boiling, **FFNO** (Tran et al., 2023), and **UNet** (Zhang et al., 2020). To add another history-aware baseline, we include **HistoryFM**, a sliding-window flow matching model that takes the full observation history as direct conditioning input and predicts each frame independently (no autoregressive feedback). Detailed descriptions of the models are in Appendix C.

**Evaluation Metrics.** In addition to standard pixelwise metrics, we also evaluate physics-informed quantities: velocity divergence (mass conservation), interface temperature, wall heat flux, and vorticity. We further decompose errors by physical region (bulk-liquid and near-interface). See Appendix E for detailed definitions.

## 5. Results and Discussion

We evaluate HB-ARFM to answer three questions: (1) Can temporal history reduce the ambiguity of inverse reconstruction under partial observability? (2) Does autoregressive conditioning preserve temporal consistency over long rollouts? (3) Are the resulting reconstructions physically meaningful and generalizable? Here, we present results for the joint temperature and velocity reconstruction (inverse task 2). Task 1 results are in Appendix F.

### 5.1. Inverse Reconstruction Under Partial Observability

In Task 2, the model must reconstruct the full temperature and velocity fields from only interface geometry and motion. While observations strongly constrain dynamics near the liquid-vapor interface, bulk transport away from the interface remains largely underdetermined.

**Snapshot Reconstruction.** Figure 3 compares representative reconstructions across baselines and Table 9 reports the quantitative metrics. Near-interface regions are recovered reasonably well by most models, where bubble geometry and velocity provide strong local constraints. However, accurately propagating these constraints into bulk transport remains challenging, where models diverge substantially.

Deterministic surrogates (FFNO and U-Net) achieve competitive average pixelwise errors but systematically suppress fine-scale structures (thermal gradients, boundary layers, and velocity wakes), producing visibly blurred reconstructions. This oversmoothing manifests in degraded spectral energy retention and suppressed velocity amplitude. History-conditioned generative models (HB-ARFM and HistoryFM) better preserve spectral energy content, as reflected in the higher HF Energy Ratio in Table 9, consistent with the sharper thermal gradients and wake structures visible in Figure 3. Flow matching recovers large-scale structure but

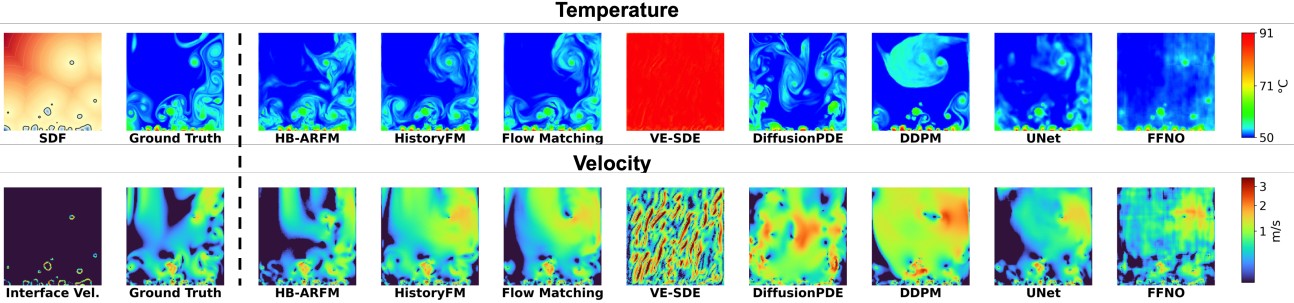

*Figure 3.* **Snapshot Reconstruction of Temperature and Velocity Fields in Subcooled Pool Boiling.** Given SDF and interface velocity as input, models predict full temperature (top) and velocity (bottom) fields. Generative methods reconstruct near-interface regions well but differ in bulk field predictions. HB-ARFM produces sharper gradients and more physically consistent reconstructions. VE-SDE exhibits catastrophic failure. Deterministic baselines (FFNO, UNet) oversmooth transport features.

shows weaker coherence in bulk regions. DiffusionPDE and DDPM generate fragmented and spatially oversmoothed fields. VE-SDE fails catastrophically, collapsing to a near-uniform state. Among all models, HB-ARFM achieves the lowest wall heat flux error, the metric most directly tied to boiling heat transfer performance.

**History-length Analysis.** We further ablate the temporal context length used by HB-ARFM by increasing the history window size $w$ from 1 to 64 past timesteps using a fixed stride of 1. Figure 4 reports the resulting temperature and velocity reconstruction errors over 30 random seeds. Increasing the history length consistently reduces reconstruction errors for both fields. The larger improvement observed for velocity reconstruction suggests that temporal history is particularly important for recovering latent flow transport dynamics.

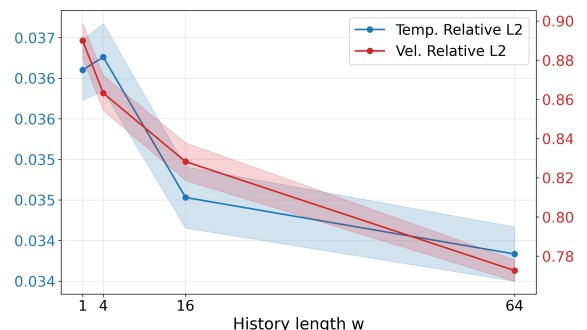

*Figure 4.* **HB-ARFM History-length Ablation.** Shaded regions denote $\pm 3\sigma$ over 30 random seeds.

### 5.2. Temporal Consistency and Rollout Reconstruction

Figure 5 evaluates rollout over ten timesteps with a stride length of 2, testing whether each model can maintain temporal consistency in the reconstructed temperature field under partial observation. HistoryFM produces visually plausible individual frames but exhibits noticeable frame-to-frame inconsistency. Wake structures behind bubbles appear and disappear stochastically, and temperature patterns drift be-

tween timesteps. It samples $\hat{X}^{(t)}$ and $\hat{X}^{(t+1)}$ from their respective conditionals independently; the model does not enforce that $\hat{X}^{(t+1)}$ is physically reachable from $\hat{X}^{(t)}$ under the governing dynamics. PDEDiff attempts to address this limitation by jointly generating a temporal window using masked conditioning over stacked target frames. However, despite training with its original randomized masking scheme to support both cold-start initialization and partially observed histories, the model fails to produce stable reconstructions and diverges from the start. DiffusionPDE both suffers from poor single-frame reconstruction and temporal inconsistency during rollout.

Deterministic surrogates fail due to the cold-start initialization problem. Bubbleformer and UNet are trained as forward predictors assuming access to the full physical state, but at inference must initialize from partial observations alone. Errors introduced during the first reconstruction step propagate through subsequent predictions, causing collapse to temporally averaged fields with suppressed dynamics and incorrect temperature ranges. These results indicate that autoregression alone cannot recover meaningful evolution without a mechanism for physically consistent inverse initialization.

In contrast, HB-ARFM maintains coherent evolution across timesteps. Thermal boundary layers evolve smoothly, condensation vortices persist behind rising bubbles, and temperature gradients propagate consistently with the underlying transport processes. By conditioning each prediction on both current sparse observations and the previously reconstructed state, HB-ARFM effectively combines inverse reconstruction and data assimilation within a unified autoregressive framework, enabling coherent multi-step prediction from partial observations.

Figure 9 further quantifies per-step rollout stability over 300 timesteps across 10 random seeds. Temperature and velocity errors remain relatively stable throughout the rollout horizon, and the variance across seeds decreases over time. Together, these result suggest that HB-ARFM's au-

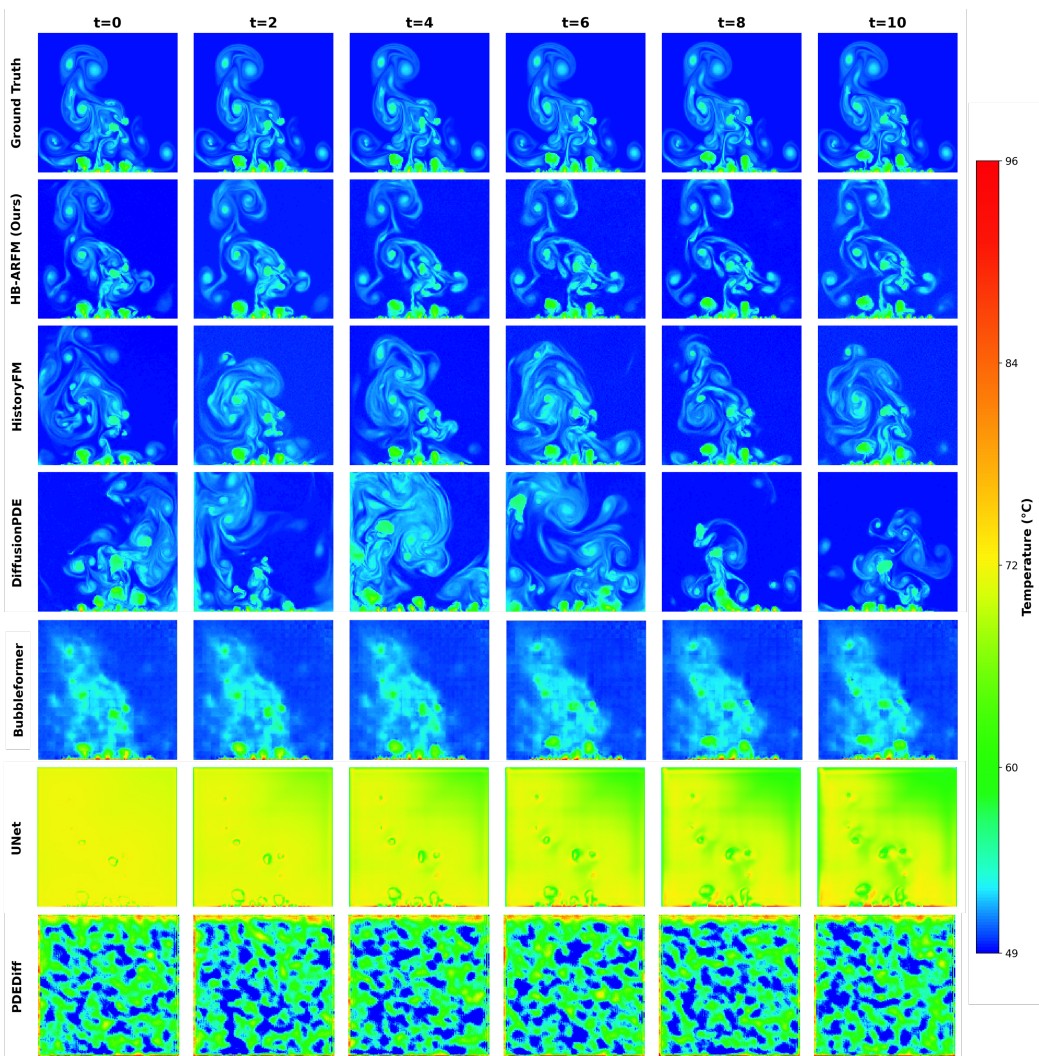

*Figure 5.* **Rollout of Reconstructed Temperature Fields in Subcooled Pool Boiling.** HB-ARFM produces temporally consistent dynamics across timesteps by combining reconstruction with data assimilation in an autoregressive framework. HistoryFM and DiffusionPDE exhibits inconsistency between timesteps. Deterministic baselines (Bubbleformer and UNet) produce blurred reconstructions or diverge immediately when initialized from partial observations.

toregressive conditioning produces temporally stable reconstructions without catastrophic error accumulation.

### 5.3. Physical Validation

**Boundary Layer and Heat Flux Reconstruction.** Heat flux quantifies the heat transfer performance associated with boiling. Critical heat flux is the heat flux at which the efficiency of cooling can drop dramatically, and boiling ceases to be an effective form of removing heat. It is arguably the most important design and safety metric for any boiling application (Liang & Mudawar, 2018).

Beyond pixelwise metrics, Figure 6 demonstrates that HB-ARFM recovers physically accurate transport. The left panel shows near-wall temperature profiles at multiple heights, demonstrating accurate boundary layer reconstruction from

heater to bulk liquid. The model captures the correct thermal gradient at the heater surface (the quantity directly determining heat flux) across the full spatial domain, not merely at isolated points like sensors and thermocouples.

**Generalization across Thermal Boundary Conditions.** The right panel evaluates generalization to unseen thermal boundary conditions, both in-distribution (ID) interpolation and out-of-distribution (OOD) extrapolation. The model achieves <2% error for ID cases and the upper extrapolation limit (117°C), with ~17% error at the lower OOD boundary (85°C) where reduced bubble nucleation yields weaker convective heat transfer. Accurate extrapolation is highly non-trivial for pool boiling due to nonlinearities. Heat flux depends on bubble nucleation dynamics, boundary layer disruption, and phase-change driven convection. The model's extrapolation suggests that it has learned the func-

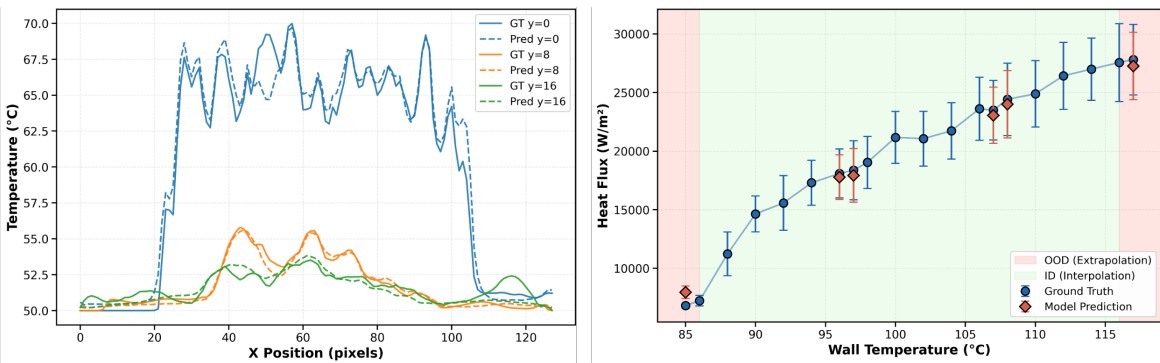

*Figure 6.* **Physics Validation of HB-ARFM. Left:** Spatial temperature profiles at different wall-normal distances ($y = 0, 8, 16$ in non-dimensional units). The $y = 0$ profile corresponds to the heated surface where the thermal boundary layer originates. The heater extends in X position from 24 to 104 in pixel space. **Right:** Wall heat flux as a function of wall temperature $T_{\text{wall}}$ evaluated over 300 timesteps. Training data spans $T_{\text{wall}} = 86\text{-}116°C$ (circles). Model performance is assessed on held-out validation temperatures include interpolation cases ($T_{\text{wall}} = 96, 97, 107, 108°C$) and extrapolation to out-of-distribution cases ($T_{\text{wall}} = 85°C$ and $117°C$, diamonds), demonstrating learned physical relationships.

tional relationship between thermal boundary conditions and phase-change heat transfer, validating its potential for reconstructions across operating conditions.

**Measurement-space Consistency.** We evaluate consistency with the observed interface measurements by comparing $H(\mathbf{x}_{\text{pred}})$, the interface velocity extracted from the predicted velocity field, against the input observations $y_{\text{obs}}$ over 300 autoregressive rollouts, as shown in Table 1. The cosine similarity of over $0.9$ indicates that the predicted velocity consistently recovers the correct interface flow direction even at long horizons. RMSE also remains stable over time. This confirms that HB-ARFM does not generate fields that contradict its conditioning.

*Table 1.* Measurement space consistency across 300 autoregressive rollouts, averaged over 5 runs.

| Horizon | RMSE | | Cosine Similarity |
|---|---|---|---|
| | **Vel-X** | **Vel-Y** | |
| 5 | $0.17 \pm 0.01$ | $0.16 \pm 0.01$ | $0.93 \pm 0.02$ |
| 50 | $0.18 \pm 0.01$ | $0.18 \pm 0.01$ | $0.93 \pm 0.01$ |
| 100 | $0.18 \pm 0.03$ | $0.18 \pm 0.02$ | $0.93 \pm 0.02$ |
| 200 | $0.19 \pm 0.02$ | $0.18 \pm 0.02$ | $0.92 \pm 0.02$ |
| 300 | $0.17 \pm 0.02$ | $0.18 \pm 0.04$ | $0.92 \pm 0.03$ |

### 5.4. Generalization to Flow Boiling

We finally evaluate whether the learned reconstruction generalizes beyond subcooled pool boiling to the substantially different *flow boiling*. Whereas pool boiling is dominated by buoyancy-driven bubble rise and localized recirculation, flow boiling introduces cross-flow advection and strong anisotropy between the streamwise and wall-normal directions. Therefore, this setting probes whether HB-ARFM learns reconstruction for a more general boiling physics.

Figure 7 shows representative reconstructions of temper-

ature and velocity fields. HB-ARFM preserves coherent thermal boundary layers near the heated wall while simultaneously reconstructing downstream wake structures induced by vapor advection. The predicted velocity fields remain aligned with the evolving interface geometry, indicating that the conditioning remains stable even in the presence of strong streamwise transport.

Quantitative results are summarized in Tables 11 and 12. HB-ARFM maintains low velocity reconstruction error in this flow-dominated boiling and an amplitude ratio of 1, indicating that the reconstructed flow preserves the correct kinetic energy scale without collapse or amplification. Importantly, the wall heat flux error remains below $0.2\%$, demonstrating accurate recovery of the near-wall thermal gradients that govern boiling heat transfer performance. Temperature reconstruction is substantially more challenging in flow boiling because thermal transport is dominated by long-range advection rather than localized buoyancy-driven circulation. Nonetheless, HB-ARFM reconstructs physically plausible thermal structures and stable rollout.

## 6. Related Work

Our work unifies probabilistic inverse reconstruction and autoregressive generative modeling to enable temporally consistent PDE state estimation under partial observations.

**Generative models for inverse PDE problems.** Diffusion and flow-based generative models have recently enabled probabilistic inverse problems in scientific computing, including inference of initial conditions, parameters, and full fields from sparse observations (Song et al., 2022; Holzschuh et al., 2023; Wang et al., 2025b). Methods such as DiffusionPDE (Huang et al., 2024) and FunDPS (Yao et al., 2025) incorporate observation guidance during sampling, yielding strong single-frame reconstructions of hid-

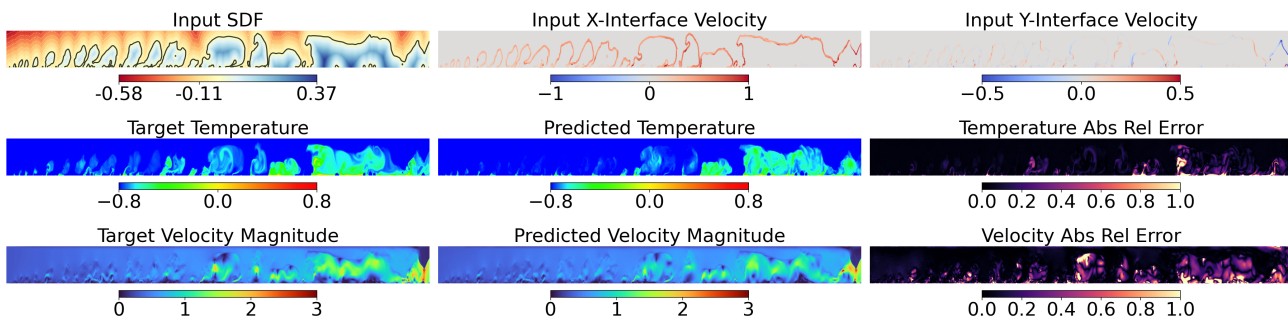

*Figure 7.* **HB-ARFM Reconstruction Results for Flow Boiling.** HB-ARFM reconstructs coupled temperature and velocity fields from sparse interface observations in flow-boiling dominated by strong streamwise advection. The model preserves near-wall thermal boundary layers, downstream wake structures, and velocity fields consistent with the evolving interface dynamics.

den fields. Others such as Denoising Diffusion Operators (Lim et al., 2025) and FunDiff (Wang et al., 2025a) extend diffusion to function spaces. In addition, several works incorporate physics-informed conditioning such as PDE residuals, conservation laws, or divergence constraints into generative objectives, training, and sampling to improve physical consistency (Shu et al., 2023; Baldan et al., 2026; Jacobsen et al., 2025; Bastek et al., 2025), extending ideas from physics-informed learning (Karniadakis et al., 2021). Despite these advances, most approaches target snapshot reconstruction, even when applied to time-dependent systems.

**Spatiotemporal dynamics reconstruction.** Recent works extend generative inverse modeling to spatiotemporal dynamics reconstruction by learning joint distributions over entire spacetime volumes. $S^3GM$ (Li et al., 2024) trains a score-based model on spatiotemporal fields and reconstructs trajectories by conditioning on sparse measurements, enforcing temporal coherence through global spacetime sampling. Rather than modeling $p(X_{0:T})$ jointly, we factorize the reconstruction over time and infer each state conditioned on past reconstructions and history of observations.

**Autoregressive flow matching and temporal generative modeling.** Autoregressive decompositions are a standard strategy for scalable sequence modeling and have been combined with diffusion models to successfully tackle forecasting and data assimilation (Shysheya et al., 2024). They factorize joint sequence distributions into tractable conditional transports, providing theoretical guarantees and efficient sampling. Autoregressive flow matching has been explored in video generation (Jin et al., 2025), motion prediction (Xie et al., 2025), and time-series forecasting (El-Gazzar & van Gerven, 2025), where conditional velocity fields evolve latent states forward in time conditioned on the generated history. Our work adopts this autoregressive factorization for inverse spatiotemporal reconstruction under partial observability. Unlike prior AR flow matching methods designed for forward generation that assume full initial states, we introduce history-bootstrapped initialization within a unified conditional transport framework, enabling latent states to be inferred from partial sparse observation sequences alone before temporal propagation. This allows us to perform causal, probabilistic reconstruction of evolving PDE fields while preserving temporal coherence.

## 7. Conclusions and Future Work

We presented HB-ARFM, a probabilistic framework for inverse PDE dynamics reconstruction under partial observability. Flow matching provides a formulation amenable to temporal composition, history bootstrapping reduces the initialization ambiguity unique to inverse reconstruction, and autoregressive conditioning maintains consistency during rollout. The result is simultaneous observation fidelity, physical plausibility, and temporal stability of full boiling dynamics reconstruction from imaging alone. The tight coupling between momentum and energy transport in boiling, mediated by moving interfaces and phase change, represents a particularly challenging benchmark. Success here suggests that HB-ARFM will generalize to other coupled PDE systems in scientific inverse problems.

**Limitations and future work.** HB-ARFM learns from data and does not explicitly enforce conservation laws or PDE constraints during training. Divergence errors indicate imperfect mass conservation. Incorporating projection steps or divergence penalties could improve conservation properties.

While increasing the history length in our models leads to reductions in error, consistent with the MZ formalism, our current history encoder remains a lossy representation of the strong predictive signal contained in the history. Addressing this limitation through more expressive history-encoding mechanisms is an important direction for future work.

Finally, all evaluations in this work are conducted on simulation data. Extending inverse reconstruction to real experimental boiling images remains challenging due to the sim-to-real gap in multiphase flows and the lack of ground-truth fields. We view robust validation on experimental data as an important open problem for future work.

## Acknowledgements

This work was supported by the Multidisciplinary University Research Initiative (MURI) program by the Office of Naval Research (ONR) under Grant No. N000142412575. We also sincerely thank the Research Cyberinfrastructure Center at the University of California Irvine, for the GPU computing resources on the HPC3 cluster.

## Impact Statement

While this work focuses on PDE-governed boiling systems, the proposed approach applies more broadly to partially observed dynamical systems, where temporal evolution provides constraining information for inverse reconstruction. Potential applications include reconstructing neural dynamics from calcium imaging, estimating atmospheric states from satellite observations, and recovering hidden dynamics in other scientific and engineering systems.

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

## A. Software and Data

Code, trained models, and evaluation scripts are available at: `https://github.com/therml-ai/HB-ARFM`.

## B. Notation

*Table 2.* Notation summary for generative models

| Symbol | Description |
|--------|-------------|
| $X_t$ | Full physical state at time $t$ |
| $\hat{X}_t$ | Reconstructed state at time $t$ |
| $y_t$ | Partial observation at time $t$ |
| $\mathcal{H}$ | Observation operator |
| $\mathbf{x}$ | Latent variable for flow matching |
| $s$ | Continuous flow time ($s \in [0, 1]$) |
| $t$ | Discrete physical timestep |
| $v_\theta$ | Learned conditional velocity field |
| $\mathbf{c}_t$ | Conditioning context at time $t$ |
| $\zeta_\phi$ | History encoder network |
| $w$ | Observation history window size |

*Table 3.* Notation summary for physical quantities

| Symbol | Description |
|--------|-------------|
| $\Omega$ | Computational domain |
| $\mathbb{T} = [0, T]$ | Time window of simulation |
| $\mathbf{u} : \mathbb{T} \times \Omega \to \mathbb{R}^2$ | Velocity field |
| $\tau : \mathbb{T} \times \Omega \to \mathbb{R}$ | Temperature field |
| $\tau_w \in \mathbb{R}$ | Temperature of the heater |
| $\phi : \mathbb{T} \times \Omega \to \mathbb{R}$ | Signed-distance function from liquid-vapor interface |
| $P : \mathbb{T} \times \Omega \to \mathbb{R}$ | Pressure field |
| $\dot{m} : \mathbb{T} \times \Omega \to \mathbb{R}$ | Evaporative mass flux |
| $\mathbf{n}_\Gamma = \nabla\phi / \lVert \nabla\phi(\cdot) \rVert_2$ | Liquid-vapor interface surface normal |
| $\Gamma = \Gamma_{\dot{m}} \subseteq \mathbb{T} \times \Omega$ | Diffused liquid-vapor interface where evaporative mass flux is non-zero |
| $\Gamma(t) \subseteq \Omega$ | The liquid-vapor interface at time $t$ |
| $\overline{\Gamma_{\dot{m}}} \subseteq \mathbb{T} \times \Omega$ | Points excluding the diffused interface |
| $q''_w : X \times \mathbb{T} \to \mathbb{R}$ | Heat flux above the heater surface |

## C. Baseline Models

We summarize the baseline models considered for comparison.

**1. Denoising Diffusion Probabilistic Models (DDPM)**   DDPM (Ho et al., 2020) defines a forward noising process and learns to reverse it by predicting Gaussian noise at each timestep. During training, a clean sample $x_0$ is progressively corrupted by noise $\epsilon$ according to a fixed schedule, producing noisy versions $x_t$ at different timesteps. A neural network is trained to predict the noise $\epsilon_\theta$ added at each timestep. At generation time, sampling starts from pure Gaussian noise $x_T$ and repeatedly applies the learned denoising steps in reverse order, gradually transforming noise into a structured sample. DDPM uses pixel space.

In our conditional setting, noise is added only to the target field (e.g., temperature), while geometric information such as the SDF is provided as an additional, noise-free input to guide the denoising process. We train a conditional DDPM where Gaussian noise is progressively added to the ground-truth field over 1000 timesteps according to a noise schedule, and a UNet is trained to predict and remove this noise. The UNet takes a multichannel input: the concatenation of the noisy target

field(s) and the conditions, along with a sinusoidal timestep embedding that tells the network how much noise is present, and outputs the predicted noise (1 channel). The UNet architecture consists of a 3-level encoder-decoder with skip connections, residual blocks with GroupNorm and SiLU activations.

**2. Variance Exploding Score-Based Models (VE-SDE).** VE-SDE is a score-based generative baseline (Song et al., 2021). Score-based models learn the score function $\nabla_x \log p_t(x)$, the gradient of the log-density of the noisy data at each noise level. Training perturbs clean data with a forward stochastic differential equation (SDE) and matches the network output to the analytically known score of the perturbed distribution. Sampling starts from Gaussian noise and numerically integrates the reverse-time SDE, using the learned score to guide samples back toward the data distribution; DDPMs are a discrete-time special case of this continuous-time framework.

In our implementation we use the Variance Exploding SDE: the forward process is $x(\sigma) = x + \sigma z$ with $z \sim \mathcal{N}(0, I)$. For numerical stability we use the noise parameterization: the network $s_\theta$ is implemented by predicting the noise $\varepsilon$ and recovering the score as $\nabla_x \log p_\sigma(x) = -\varepsilon/\sigma$, with denoising score matching loss $\mathbb{E}[\|\varepsilon_\theta(x + \sigma z, \sigma) - z\|^2]$ (equivalent to matching the score). The backbone is an NCSN++-style U-Net with $\sigma$-conditioned residual blocks (sinusoidal embeddings), group normalization. Sampling integrates the reverse-time SDE via predictor–corrector steps (reverse diffusion plus Langevin corrector).

**3. Flow Matching.** Flow Matching (Lipman et al., 2023) is a framework for learning continuous-time generative models that transport samples from a simple base distribution to a target distribution by integrating a learned velocity field. Given conditioning information $\mathbf{c}$, flow matching learns a time-dependent velocity field $v_\theta(\mathbf{x}, \mathbf{c}, s)$ defining the ordinary differential equation $d\mathbf{x}(s)/ds = v_\theta(\mathbf{x}(s), \mathbf{c}, s)$ for $s \in [0, 1]$, with initial condition $\mathbf{x}(0) = \mathbf{x}^0 \sim \mathcal{N}(\mathbf{0}, \mathbf{I})$. Solving this ODE yields $\mathbf{x}(1)$, a sample from the learned conditional distribution. The velocity field is trained via a regression objective that matches the instantaneous velocity of a prescribed probability path between $\mathbf{x}^0$ and the target $X$.

In our implementation we use Optimal Transport Conditional Flow Matching (OT-CFM): the training path is $x(s) = s \cdot x_1 + (1-s) \cdot x_0$ (linear interpolation) with target velocity $v_{\text{target}} = x_1 - x_0$, and the network predicts $v_\theta(x_s, \text{condition}, s)$ with MSE loss $\mathbb{E}[\|v_\theta(x_s, \mathbf{c}, s) - v_{\text{target}}\|^2]$ where $t \sim \mathcal{U}(0, 1)$. The backbone is a 3-level U-Net with time-conditioned residual blocks, group normalization, and optional bottleneck attention. Sampling integrates the ODE from Gaussian noise using Heun.

**4. Bubbleformer.** Bubbleformer (Hassan et al., 2025) is a spatiotemporal transformer-based neural surrogate for multiphase boiling simulations that can act as a robust forecasting model for a boiling system. It uses spatial and temporal attention to capture long-range spatial correlations and temporal dependencies arising from phase-change dynamics. The model is trained autoregressively on fixed-length history windows to forecast future states of the system.

**5. FFNO.** FFNO (Tran et al., 2023) is a 2D Factorized Fourier Neural Operator: it applies learnable spectral convolutions in the Fourier domain along each spatial dimension separately, retaining a fixed number of low-frequency modes per dimension, then passes the result through small feedforward networks. Our implementation uses shared Fourier weights and shared feedforward networks across spectral layers for parameter efficiency, along with weight normalization and layer normalization for stability.

**6. DiffusionPDE.** DiffusionPDE (Huang et al., 2024) is a conditional diffusion baseline for PDE field prediction under partial observation. It uses the EDM (Karras et al., 2022) framework: a denoising network is trained to predict the clean target given noisy target and conditioning, with input/output preconditioning (scale and skip terms depending on noise level $\sigma$) and log-normal sampling of $\sigma$ during training. The backbone is a SongUNet (DDPM++/NCSN++ style) with timestep embeddings, group normalization, and self-attention at selected resolutions. Training minimizes a weighted MSE loss with EDM's $\sigma$-dependent weighting; inference uses a discrete schedule in $\sigma$ with a Heun ODE solver.

**7. UNet.** UNet (Ronneberger et al., 2015) is a standard convolutional encoder–decoder baseline. The 2D UNet uses a four-level encoder with repeated blocks of two 3×3 convolutions, batch normalization, and GELU activation, downsampling via max pooling; a bottleneck at the deepest level; and a symmetric decoder with transposed convolutions for upsampling and skip connections from the encoder. A 1×1 convolution produces the final channel-wise outputs.

**8. PDEDiff.** PDEDiff (Shysheya et al., 2024) is a conditional diffusion model that generates a full temporal window of $w$ frames jointly rather than predicting one frame at a time. The window is flattened along the channel axis, $X \in \mathbb{R}^{w \cdot C_{\text{out}} \times H \times W}$, and the model is trained with an amortized mask-based conditioning scheme: the input to the score network is the concatenation $[\mu, \mu \odot X, \mathbf{c}]$, where $\mu \in \{0, 1\}^{w \cdot C_{\text{out}} \times H \times W}$ is a binary mask indicating which target channels are observed, $\mu \odot X$ provides their values, and $\mathbf{c}$ is the conditioning window (SDF + interface velocity) stacked along channels. During training, the number of revealed past frames is randomized so the same network handles cold-start prediction (no observed history), partially observed histories, and fully observed histories, all amortized in one model. Diffusion uses a Variance-Preserving SDE with a cosine $\alpha$ schedule, $x_s = \sqrt{\alpha(s)}\, x_0 + \sqrt{1 - \alpha(s) + \eta^2}\, \epsilon$, and noise-prediction parameterization $\epsilon_\theta(x_s, \mathbf{cond}, s)$ trained with MSE; sampling integrates the reverse SDE with optional Langevin corrector steps and uses an EMA copy of the network. The backbone is the original PDEDiff UNet: a multi-resolution encoder-decoder of context-residual blocks where the sinusoidal time embedding is injected as an additive bias (rather than via FiLM/group-norm scale-shift), with channel-last LayerNorm and nearest-neighbour upsampling. At inference, we use a sliding-window autoregressive rollout: at each step the model generates the full window jointly, the past $w - h$ frames are filled with previously predicted values via the mask, and the last $h$ predicted frames advance the trajectory.

**9. HistoryFM.** History-window Flow Matching is a history-conditioned extension of OT-CFM (described above in §3) that augments the per-frame conditioning with a sliding window of $w$ past observable frames, providing the model with explicit temporal context while remaining non-autoregressive. Given a window of past conditioning frames $\{(\text{SDF}_{t-w+1}, \mathbf{u}_{t-w+1}^\Gamma), \dots, (\text{SDF}_t, \mathbf{u}_t^\Gamma)\}$, the frames are flattened along the channel axis into a single tensor $\mathbf{c} \in \mathbb{R}^{w \cdot C_{\text{cond}} \times H \times W}$ and concatenated with the noisy target $x_0$ as input to the velocity network $v_\theta(x_s, \mathbf{c}, s)$.

# D. Datasets and Training

### D.1. Datasets

To evaluate generalization across different boiling types, we consider two subsets from the BubbleML 2.0 dataset (Hassan et al., 2025).

**Pool Boiling.** We study subcooled pool boiling of FC-72, a dielectric that finds application in immersion cooling of electronic components. The simulations are performed on a 2D computational domain $\Omega \subset \mathbb{R}^2$ with spatial co-ordinates $(x, y)$ discretized on a uniform Cartesian grid. The domain spans $x \in [-8, 8]$ and $y \in [0, 16]$ in non-dimensional units, with grid spacing $\Delta x = \Delta y = 1/32$. The corresponding lengths in physical units can be obtained by multiplying the characteristic length $l_c = 0.73$ millimeters for FC-72 by the corresponding non-dimensional values. A constant temperature 1D heater is located at the bottom of the domain, spanning $x \in [-5.25, 5.25]$. The dataset consists of 20 different trajectories at wall temperatures ranging from $85°C$ to $117°C$. We use 14 of these ($T_{wall} \in \{86, 88, 90, 92, 94, 98, 100, 102, 104, 106, 110, 112, 114, 116\}°C$) for training, and 6 are left out for testing. Among the test set, 4 ($T_{wall} \in \{96, 97, 107, 108\}°C$) are in-distribution, and 2 ($T_{wall} \in \{85, 117\}°C$) are out-of-distribution.

**Flow Boiling.** We additionally consider subcooled flow boiling of FC-72 based on experiments conducted at NASA Glenn Research center (Konishi et al., 2015a;b) at microgravity and simulated using Flash-X (Dubey et al., 2022). We chose these cases because the dataset has clearly segmentable bubbles (Chang et al., 2023) compatible with our computer vision pipeline. Compared to pool boiling, the physical domain is significantly larger ($\sim 115mm \times 5mm$) and is also highly anisotropic. Adaptive mesh refinement is used in the original simulations, with the grid non-dimensionalized to ($\sim 161 \times 7$). The block-structured AMR mesh is then interpolated to a regular $5152 \times 224$ grid, preserving the same grid spacing used in the pool boiling simulations ($\Delta x = \Delta y = 1/32$). Similar to pool boiling, there exists a 1D heater at the bottom of the domain. In this case, the heater spans the entire x-axis and a constant heat flux boundary condition results in trajectories with different applied wall heat fluxes ($q_{wall} \in \{10, 15, 20, 25, 30, 40, 50, 60, 70\}\%\text{CHF}$). The $15\%\text{CHF}$ trajectory is left out for testing and others are used in training.

### D.2. Training

For efficient training, we downsample the datasets using bilinear interpolation. The downsampling factor is set to 4 for the pool boiling datasets and 2 for the flow boiling datasets. All models are trained for 25 epochs with a batch size of 16. We use the AdamW optimizer with a cosine learning-rate schedule, 1000 warmup iterations, and a minimum learning rate of $10^{-6}$. The main training hyperparameters are summarized in Table 4, while the model configuration is provided in Table 5.

*Table 4.* Training hyperparameters used for model optimization.

| Hyperparameter | Value |
|---|---|
| Pool boiling downsampling factor | 4 |
| Flow boiling downsampling factor | 2 |
| Interpolation method | Bilinear |
| Number of epochs | 25 |
| Batch size | 16 |
| Optimizer | AdamW |
| Learning rate | $2.5 \times 10^{-4}$ |
| Weight decay | $1.0 \times 10^{-2}$ |
| Learning-rate scheduler | Cosine schedule with warmup |
| Warmup iterations | 1000 |
| Minimum learning rate | $1.0 \times 10^{-6}$ |

*Table 5.* Model configuration for the HB-ARFM model.

| Parameter | Value |
|---|---|
| **Flow Matching backbone** | |
| UNet Blocks | 3 |
| Hidden channels | $[32, 64, 128]$ |
| Flow Matching time embedding dimension | 256 |
| Autoregressive rollout length | 5 |
| Attention in UNet backbone | No |
| **History Encoder** | |
| History length | 64 |
| History stride | 1 |
| Hidden channels | 32 |
| Encoder blocks | 2 |
| Attention embedding dimension | 128 |
| Number of heads | 8 |
| Patch size | 8 |
| MLP ratio | 4.0 |
| **Loss and inference configuration** | |
| Bootstrap loss weight | 1.0 |
| Autoregressive loss weight | 1.0 |
| Inference ODE solver | Heun |
| Number of integration steps | 50 |

# E. Evaluation Metrics

We evaluate our model using metrics listed in Table 6 that assess both the accuracy of predicted fields and their physical plausibility. These metrics are designed specifically for two-phase flow simulations and are organized into six groups: (1) global accuracy metrics, (2) region-wise error metrics, (3) conservation metrics, (4) thermal metrics, (5) kinematic metrics, and (5) statistical / spectral metrics.

In this section, we use discretizations of space $\Omega$ and time $\mathbb{T} = [0, T]$: space is discretized into $n$ points $\Omega_n$ and time is discretized into $k$ points $\mathbb{T}_k$. These are both discretized on a uniform cartesian grid so we can apply finite differences. All spatial derivatives are computed using second-order central finite differences; for boundary cells, one-sided differences are used.

We compute metrics for the velocity $\mathbf{u}$, temperature $\tau$, and signed distance $\phi$ fields and compare them with the ground truth simulation. For metrics that are generic and applicable to any field, we will write a generic field $f \in \{\mathbf{u}, \tau, \phi\}$. We use $f$ to represent the ground truth field, and $\hat{f}$ as the model's output. Recall that $f : \Omega \times \tau \to \mathbb{R}^d$ and that $d = 2$ for velocity and $d = 1$ for temperature and the signed distance field.

*Table 6.* Summary of the evaluation metrics.

| Category | Metric | Physical Significance |
|---|---|---|
| Global Accuracy | Rel. L2 | Aggregate relative error |
| | Max Rel. L2 | Worst-frame relative error |
| | $\|\cdot\|_\infty$ | Pointwise tail error |
| Region-Wise Error | IRMSE | Interface-region RMSE |
| | BRMSE | Bulk-liquid RMSE |
| Conservation | $\text{RMSE}(\nabla \cdot \mathbf{u})$ | Mass conservation |
| Thermal | $\bar{q}_w''$ | Wall heat transfer rate |
| | $\text{RE}_{q''}$ | Heat flux prediction accuracy |
| Kinematic | $\text{RMSE}(\omega)$ | Rotational structure of the flow |
| Statistical / Spectral | $A_\mathbf{u}$ | Velocity amplitude bias |
| | $R_{\text{HF}}(\tau)$ | High-wavenumber thermal detail |

## E.1. Global Accuracy Metrics

**Relative L2 Error.** $\text{RelL2}(f, \hat{f}) = ||f - \hat{f}||_2 / ||f||_2$.

**Maximum Relative L2 Error.** The worst single-frame relative L2 across the evaluation window,

$$\text{MaxRelL2}(f, \hat{f}) = \max_t \ \text{RelL2}(f(t, \cdot), \hat{f}(t, \cdot)) \tag{3}$$

This metric can identify cases where the mean error appears acceptable, but the model performs poorly on individual frames. This is a common failure mode of stochastic samplers.

**Maximum Error.** The pointwise $L^\infty$ error, $\text{MaxError}(f, \hat{f}) = \max \left| f - \hat{f} \right|$, penalizes rare, but large, errors that may be averaged out by the L2 norms.

## E.2. Region-wise Error Analysis

To provide spatially-resolved error analysis, we partition the domain into physically meaningful regions and report the RMSE of each field restricted to those regions. These metrics rely on tracking the location of the sharp liquid-vapor interface ($\Gamma$). The signed distance function $\phi(\mathbf{x}, t)$ defines the sharp liquid-vapor interface, with the sign specifying the phase:

$$\phi(\mathbf{x}, t) \begin{cases} < 0 & \text{liquid phase} \\ = 0 & \text{immersed boundary (bubble boundary)} \\ > 0 & \text{vapor phase (inside bubble)} \end{cases} \tag{4}$$

However, considering the zero level-set as the sharp interface does not take into account the effects of evaporative mass transfer that occur in the vicinity. Thus, we use a diffused interface region defined by the mass flux for our interfacial metrics.

**Diffused Interface.** The diffused interface ($\Gamma_{\dot{m}}$) is defined as the region in the domain where mass flux ($\dot{m}$) is non-zero. The continuity equation for multiphase systems is given by $\nabla \cdot \mathbf{u} = -\dot{m}\mathbf{n}_\Gamma \cdot \nabla \left(\frac{1}{\rho'}\right)\Big|_\Gamma$, where $\mathbf{n}_\Gamma$ is a unit vector normal to the liquid-vapor interface.

**Bulk Liquid.** We define the bulk liquid region as the region where incompressibility holds. The bulk liquid is the set of cells inside the liquid phase, where $\phi < 0$, that are sufficiently far from the interface and the heated wall:

$$\mathcal{B} = \left\{ t \in \mathbb{T}_k, \mathbf{x} \in \Omega_n \ : \ \phi(t, \mathbf{x}) < 0 \ \wedge \ \mathbf{x} \notin \Gamma_{\dot{m}}(t) \ \wedge \ j \geq j_{\text{wall}} \right\}, \tag{5}$$

with $j_{\text{wall}} = 16$ at the native resolution (corresponding to $j \geq 4$ rows with a downsampling factor of 4). Excluding the wall region prevents the strongly forced thermal boundary layer from dominating bulk-liquid statistics.

**Region-wise RMSE.** For a field $f \in \{\mathbf{u}, \tau\}$ and region $\mathcal{R} \in \{\Gamma_{\dot{m}}, \mathcal{B}\}$, we compute the RMSE restricted to that region:

$$\text{RMSE}_{\mathcal{R}}(f, \hat{f}) = \sqrt{\frac{1}{|\mathcal{R}|} \sum_{(t,\mathbf{x}) \in \mathcal{R}} \left(f(t,\mathbf{x}) - \hat{f}(t,\mathbf{x})\right)^2}. \tag{6}$$

We refer to the interface RMSE as IRMSE and the bulk-liquid RMSE as BRMSE. For velocity, the integrand is the squared magnitude of the component-wise vector errors so that errors in both components are aggregated. This region-wise decomposition allows us to identify which physical regions present the greatest modeling challenges and provides insight into the model's behavior in different flow regimes (interfacial dynamics versus bulk advection).

### E.3. Velocity Divergence (Mass Conservation)

In the region where mass flux is zero (away from the interface), the continuity equation boils down to the single phase continuity equation for incompressible fluids, $\nabla \cdot \mathbf{u} = 0$. Thus, mass conservation implies that the velocity field is divergence-free. We compute the divergence using central finite differences:

$$\nabla \cdot \mathbf{u} = \frac{\partial u^x}{\partial x} + \frac{\partial u^y}{\partial y} \approx \frac{u^x_{i+1,j} - u^x_{i-1,j}}{2\Delta x} + \frac{u^y_{i,j+1} - u^y_{i,j-1}}{2\Delta y} = D \tag{7}$$

where $u^x$ and $u^y$ are the $x$- and $y$-components of velocity, respectively.

Rather than measuring the residual divergence of the ground truth and predicted field in isolation, we compare the model's divergence *against* the divergence of the reference simulation. We report the RMSE of the divergence error over non-interface cells:

$$\text{RMSE}(D, \hat{D}) = \sqrt{\frac{1}{|\overline{\Gamma_{\dot{m}}}|} \sum_{(t,\mathbf{x}) \in \overline{\Gamma_{\dot{m}}}} \left(D - \hat{D}\right)^2}. \tag{8}$$

### E.4. Wall Heat Flux

The wall heat flux quantifies the rate of heat transfer from the heated surface to the fluid, which is the primary quantity of interest in boiling heat transfer applications. We compute the heat flux using Fourier's law at points along the bottom at $y = 0$, which corresponds to a distance of $\frac{\Delta y}{2}$ over the heater surface as:

$$q''_w(x, t) = k_l \frac{\tau_w - \tau((x, \frac{\Delta y}{2}), t)}{0.5 \, \Delta y \, l_c} \tag{9}$$

where $k_l = 6.25 \times 10^{-2}$ W/(m·K) is the thermal conductivity of liquid FC-72, $l_c = 0.73 \times 10^{-3}$ m is the characteristic length scale used to non-dimensionalize the simulation, and $\tau_{\text{w}}$ is the heater temperature. The heat flux is computed only for cells in the liquid phase, where $\phi < 0$, and above the heater, at the points $x \in [-5.25, 5.25]$.

The spatially-averaged heat flux at each timestep is the mean over the heater-region wall-row cells,

$$\bar{q}''_w(t) = \frac{1}{|X_h|} \sum_{x \in X_h} q''_w(x, t), \qquad X_h = \{x : x \in [-5.25, \, 5.25]\}, \tag{10}$$

We report the temporal mean and the relative error against ground truth:

$$\text{RE}_{q''} = \frac{\left\| \bar{q}''_{w,\text{pred}} - \bar{q}''_{w,\text{GT}} \right\|_2}{\left\| \bar{q}''_{w,\text{GT}} \right\|_2} \times 100\% = \frac{\sqrt{\sum_{t \in \mathbb{T}_k} \left(\bar{q}''_{w,\text{pred}}(t) - \bar{q}''_{w,\text{GT}}(t)\right)^2}}{\sqrt{\sum_{t \in \mathbb{T}_k} \bar{q}''_{w,\text{GT}}(t)^2}} \times 100\%. \tag{11}$$

Because this only depends on the wall-row temperature and the heater-region, it tightly couples temperature accuracy and interface localization, and is therefore a sensitive end-to-end measure of physical realism.

### E.5. Vorticity

Vorticity measures the local rotation of the fluid and is particularly important for capturing the swirling flow patterns around rising bubbles. In 2D, the vorticity is the $z$-component of the curl of velocity:

$$\frac{\partial u^y}{\partial x} - \frac{\partial u^x}{\partial y} \approx \frac{u^y_{i+1,j} - u^y_{i-1,j}}{2\Delta x} - \frac{u^x_{i,j+1} - u^x_{i,j-1}}{2\Delta y} = \omega. \tag{12}$$

Positive vorticity indicates counter-clockwise rotation, while negative vorticity indicates clockwise rotation. High vorticity magnitudes are characteristic of the wake regions behind rising bubbles. To assess whether a model has learned the correct rotational structure of the flow rather than a smoothed mean field, we report the RMSE of the vorticity error between the prediction and ground truth, restricted to non-interface cells:

$$\mathrm{RMSE}(\omega, \hat{\omega}) = \sqrt{\frac{1}{|\Gamma_{\dot{m}}|} \sum_{(t, \mathbf{x}) \in \overline{\Gamma_{\dot{m}}}} (\omega - \hat{\omega})^2}. \tag{13}$$

Interface cells are excluded for the same reason as in the divergence metric: the vorticity across the diffused interface would otherwise dominate the score due to the effects of evaporative mass flux.

### E.6. Statistical and Spectral Metrics

The two metrics in this group complement the pointwise errors of the previous sections by quantifying biases in the predicted fields: amplitude collapse for velocity and spectral smoothing for temperature. Both are sign-aware, so the ideal value is 1, and values below or above 1 indicate different types of errors. These expose model behavior that norm-based metrics may not capture.

**Velocity Amplitude.** A common failure mode of regression-based and diffusion-based predictors is amplitude collapse: the predicted flow has roughly the right structure but is systematically too weak. I.e., the predicted velocity field may tend to predict slower velocities. To detect this, we compare the spatially averaged speed per frame and then compute a time average of the ratio of the predicted and ground truth speeds.

$$\bar{u}_{\mathrm{GT}}(t) = \frac{1}{|\Omega_n|} \sum_{\mathbf{x} \in \Omega_n} \|\mathbf{u}_{\mathrm{GT}}(t, \mathbf{x})\|, \tag{14}$$

$$\bar{u}_{\mathrm{pred}}(t) = \frac{1}{|\Omega_n|} \sum_{\mathbf{x} \in \Omega_n} \|\mathbf{u}_{\mathrm{pred}}(t, \mathbf{x})\|, \tag{15}$$

$$A_{\mathbf{u}} = \frac{1}{k} \sum_{t \in \mathbb{T}_k} \frac{\bar{u}_{\mathrm{pred}}(t)}{\bar{u}_{\mathrm{GT}}(t)}. \tag{16}$$

The ideal value is $A_{\mathbf{u}} = 1$. $A_{\mathbf{u}} < 1$ indicates systematic damping of the predicted velocity magnitude, while $A_{\mathbf{u}} > 1$ indicates over-amplification. This metric is intended to capture amplitude bias in the speed field; it does not by itself assess directional accuracy, spatial localization, or kinetic energy, and is therefore reported alongside standard vector-field error metrics.

**Temperature High-Frequency Energy.** To quantify whether a model preserves fine-scale thermal structure, we compare the fraction of spectral energy carried by high wavenumbers in the prediction and ground truth. For each frame $t$, we compute the 2D power spectrum $P$ and its high-frequency fraction. Here $\kappa_x$ and $\kappa_y$ are the non-negative integer DFT bin indices in the $x$ and $y$ directions (wave numbers). $\hat{\tau}$ is the discrete Fourier transform of $\tau$.

$$P_t(\kappa_x, \kappa_y) = \left| \hat{\tau}_t(\kappa_x, \kappa_y) \right|^2, \qquad \kappa_r = \sqrt{\kappa_x^2 + \kappa_y^2}, \tag{17}$$

$$\mathrm{HF}_t = \frac{\sum_{\kappa_r \geq \kappa_{\mathrm{HF}}} P_t(\kappa_x, \kappa_y)}{\sum_{\kappa_x, \kappa_y} P_t(\kappa_x, \kappa_y)}, \qquad \kappa_{\mathrm{HF}} = 12. \tag{18}$$

The threshold $k_{\mathrm{HF}} = 12$ separates large-scale structure (bulk thermal plumes, big vortices) from small-scale detail (thin thermal boundary layers, fine gradients across the diffused interface, small eddies). The reported ratio is

$$R_{\mathrm{HF}}(\tau) = \frac{\frac{1}{N_t} \sum_t \mathrm{HF}_t^{\mathrm{pred}}}{\frac{1}{N_t} \sum_t \mathrm{HF}_t^{\mathrm{GT}}}. \tag{19}$$

A value $R_{\mathrm{HF}} < 1$ indicates spectral smoothing (the most common failure mode of regression-based and overly-conditioned diffusion models); a value $R_{\mathrm{HF}} > 1$ indicates noise or ringing artifacts in the predicted field.

## F. Additional Results

### F.1. Task 1 Inverse Reconstruction

**Snapshot Reconstruction.** Figure 8 compares temperature reconstruction from interface geometry alone and Table 7 summarizes the metrics over 50 random snapshots. In contrast to Task 2, where both interface geometry and interface velocity are available, Task 1 requires recovering the thermal field from the signed-distance function only. This substantially increases the ambiguity of the inverse problem because the interface geometry contains limited information about the underlying bulk transport and advective thermal structure away from the vapor boundary.

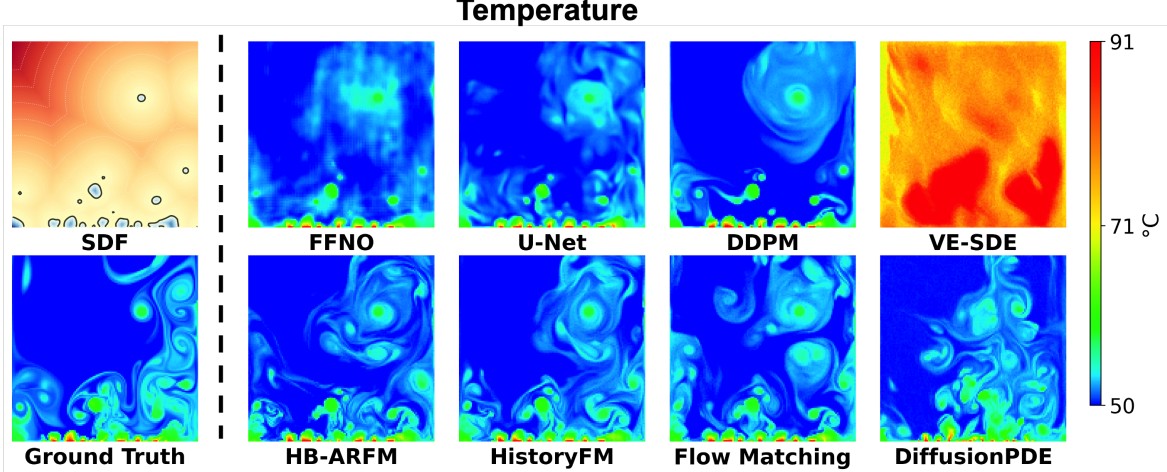

*Figure 8.* **Temperature Prediction from SDF (Task 1) in Subcooled Pool Boiling.** Given the signed distance function describing bubble position, models reconstruct the temperature field. The colorbar indicates temperature in °C.

*Table 7.* Results for temperature reconstruction (task 1) in subcooled pool boiling over 50 random snapshots. **Bold** denotes the best among diffusion models, *italic* denotes the best among all models.

| Metric | Stochastic Models | | | | | | Deterministic Models | |
|---|---|---|---|---|---|---|---|---|
| | HB-ARFM | Flow Matching | HistoryFM | VE-SDE | DiffusionPDE | DDPM | UNet | FFNO |
| Temp. Relative L2 | 0.037 | 0.038 | 0.036 | 0.566 | 0.057 | **0.035** | *0.032* | 0.033 |
| Temp. Max Rel L2 | 0.041 | 0.044 | **0.040** | 0.605 | 0.071 | 0.041 | *0.037* | *0.037* |
| Temp. Max Error (°C) | **29.887** | 31.676 | 30.837 | 55.040 | 42.974 | 40.237 | *22.561* | 28.609 |
| Temp. IRMSE | 3.623 | 2.548 | **2.525** | 31.441 | 7.025 | 3.099 | *2.173* | 2.543 |
| Temp. BRMSE | 1.786 | 1.888 | 1.757 | 29.384 | 2.292 | **1.651** | *1.598* | 1.615 |
| Temp. HF Energy Ratio | 0.962 | *0.968* | 0.956 | 2.562 | 0.881 | 0.954 | 0.801 | 0.877 |
| Wall Heat Flux Rel. Error (%) | 4.632 | *3.261* | 3.705 | 54.171 | 18.458 | 3.369 | 4.377 | 3.485 |

While history-conditioned flow matching models remain competitive overall, the advantage of temporal history is less pronounced when conditioning only on interface geometry. HB-ARFM, HistoryFM, and snapshot Flow Matching all recover similar large-scale thermal plume structures and near-interface gradients, suggesting that the absence of interface velocity observations removes much of the temporal information needed to constrain the latent transport dynamics.

Nevertheless, clear differences remain between model families. The flow matching models produce the most physically coherent reconstructions overall, preserving sharp thermal layers near the heater while maintaining smooth advective structures in the bulk liquid. Correspondingly, flow matching attains the best heat flux error and highest high-frequency energy ratio. DDPM performs surprisingly well near the liquid-vapor interface and reconstructs localized thermal boundary layers with relatively accurate wall heat flux. However, the bulk liquid region remains visibly over-smoothed, especially in the upper-right corner of the domain where long-range transport dominates. DiffusionPDE exhibits the opposite failure mode. While the reconstructed bulk field retains sharper large-scale transport structure than DDPM, the near-wall thermal gradients are poorly reconstructed, leading to visibly incorrect heater boundary layers and substantially larger wall heat flux error. VE-SDE fails entirely in this setting similar to task 2. The deterministic baselines (UNet and FFNO) achieve competitive pixelwise errors but do so through overly smooth predictions that suppress physically important transport structure. These results indicate that global reconstruction error alone is insufficient for evaluating inverse boiling reconstruction.

**Temporal Reconstruction and Rollout.** Table 8 evaluates HB-ARFM under autoregressive rollout over increasing horizons. Despite the limited observability of Task 1, the model remains stable across long prediction windows, with relative $\ell_2$ error remaining nearly constant through 300 rollout steps. Importantly, both interface RMSE and the wall heat flux error remain relatively stable, indicating that the autoregressive conditioning avoids the accumulation of catastrophic thermal drift over long horizons.

*Table 8.* Results for temperature reconstruction (task 1) in subcooled pool boiling across different rollout steps.

| Metric | 1 step | 100 steps | 200 steps | 300 steps |
|---|---|---|---|---|
| Temp. Relative L2 | $0.040 \pm 0.002$ | $0.037 \pm 0.000$ | $0.037 \pm 0.000$ | $0.038 \pm 0.000$ |
| Temp. Max Error (°C) | $16.499 \pm 1.945$ | $21.556 \pm 0.394$ | $21.157 \pm 0.310$ | $21.792 \pm 0.230$ |
| Temp. IRMSE | $2.590 \pm 0.213$ | $3.543 \pm 0.049$ | $3.469 \pm 0.027$ | $3.555 \pm 0.021$ |
| Temp. BRMSE | $1.986 \pm 0.105$ | $1.787 \pm 0.013$ | $1.791 \pm 0.007$ | $1.841 \pm 0.006$ |
| Temp. HF Energy Ratio | $0.902 \pm 0.032$ | $0.915 \pm 0.010$ | $0.891 \pm 0.004$ | $0.910 \pm 0.005$ |
| Wall Heat Flux Rel. Error (%) | $3.077 \pm 1.878$ | $5.500 \pm 0.489$ | $5.169 \pm 0.258$ | $5.250 \pm 0.232$ |

### F.2. Task 2 Inverse Reconstruction

**Snapshot Reconstruction.** Table 9 summarizes the metrics for joint velocity and temperature reconstruction. HB-ARFM achieves the best or near-best performance on bulk physical quantities, including velocity and temperature BRMSE, vorticity RMSE, and wall heat flux error. However, it does not attain the lowest error on IRMSE, where methods that directly condition on instantaneous observations have an advantage.

Overall, HB-ARFM prioritizes physically consistent bulk reconstruction across scales, achieving competitive performance on global metrics while trading off some accuracy on quantities directly specified by the conditioning signal at a single snapshot. The amplitude ratio captures global energy calibration and the HF energy ratio captures spectral fidelity; strong performance on both by history-conditioned models indicates that these models preserve bulk energy levels and fine-scale structure simultaneously. The best performance of HB-ARFM on wall heat flux further indicates accurate boundary-layer behavior and energy transfer.

**Temporal Reconstruction and Rollout.** Table 10 and Figure 9 summarize the reconstruction results across 300 rollout steps.

### F.3. History Encoder Ablation

We run a direct ablation replacing the learned history encoder with (a) zero initialization and (b) mean initialization, which averages the observation history over 10 past timesteps. All three variants are identical after frame 0 and differ only in how the initial state is computed before autoregressive rollout begins. As shown in Figure 10, both zero and mean initialization degrade substantially relative to the learned encoder, validating that the encoder's ability to infer the initial bulk state from interface history is responsible for the performance gains.

*Table 9.* Results for temperature and velocity reconstruction (task 2) in subcooled pool boiling over 50 random snapshots. IRMSE refers to the liquid-vapor interface and BRMSE refers to bulk liquid. **Bold** denotes the best among diffusion models, *italic* denotes the best among all models.

| Category | Metric | Stochastic Models | | | | | | | Deterministic Models | |
|---|---|---|---|---|---|---|---|---|---|---|
| | | HB-ARFM | Flow Matching | HistoryFM | VE-SDE | PDEDiff | DiffusionPDE | DDPM | UNet | FFNO |
| Velocity Metrics | Vel. Relative L2 | 0.747 | 0.758 | **0.713** | 1.939 | 10.579 | 1.532 | 0.969 | *0.695* | 0.827 |
| | Vel. Max Rel L2 | **0.905** | 1.019 | 0.923 | 2.510 | 14.005 | 2.316 | 1.529 | *0.849* | 1.062 |
| | Vel. Max Error ($\times l_c$ ms$^{-1}$) | 3.126 | 3.463 | *3.074* | 8.175 | 16.108 | 7.399 | 4.634 | 3.171 | 3.474 |
| | Vel. IRMSE | 0.250 | **0.186** | 0.290 | 0.606 | 7.444 | 1.357 | 0.369 | *0.149* | 0.481 |
| | Vel. BRMSE | 0.537 | 0.546 | **0.510** | 1.394 | 4.979 | 1.063 | 0.696 | *0.501* | 0.586 |
| | Vel. Amplitude Ratio | 0.851 | 1.024 | *1.002* | 1.764 | 12.563 | 1.208 | 1.244 | 0.775 | 0.847 |
| | Vel. Divergence RMSE (excl. interface) | 0.191 | **0.161** | 0.192 | 0.747 | 9.177 | 0.312 | 0.175 | *0.128* | 0.457 |
| | Vorticity RMSE (excl. interface) | 1.082 | **1.059** | 1.086 | 6.127 | 11.661 | 1.964 | 1.248 | *0.969* | 1.197 |
| Temperature Metrics | Temp. Relative L2 | **0.033** | 0.034 | 0.034 | 0.761 | 0.237 | 0.061 | 0.042 | *0.030* | 0.031 |
| | Temp. Max Rel L2 | **0.038** | 0.039 | 0.040 | 0.804 | 0.254 | 0.072 | 0.059 | 0.035 | *0.033* |
| | Temp. Max Error (°C) | **28.574** | 29.562 | 34.662 | 50.920 | 100.312 | 43.652 | 45.251 | *22.636* | 33.342 |
| | Temp. IRMSE | 2.565 | **2.420** | 3.236 | 32.213 | 15.021 | 7.459 | 4.094 | *1.649* | 2.860 |
| | Temp. BRMSE | **1.632** | 1.714 | 1.640 | 39.937 | 11.900 | 2.470 | 1.971 | 1.516 | *1.440* |
| | Temp. HF Energy Ratio | 0.920 | 0.892 | 0.949 | 0.234 | 9.187 | **1.015** | 1.210 | 0.862 | 0.692 |
| | Wall Heat Flux Rel. Error (%) | *2.254* | 2.569 | 3.926 | 87.767 | 111.417 | 19.698 | 6.300 | 3.461 | 3.181 |

*Table 10.* Results for temperature and velocity reconstruction (task 2) in subcooled pool boiling across different rollout steps.

| Metric | 1 step | 100 steps | 200 steps | 300 steps |
|---|---|---|---|---|
| Vel. Relative L2 | $0.917 \pm 0.015$ | $0.938 \pm 0.004$ | $0.892 \pm 0.003$ | $0.863 \pm 0.002$ |
| Vel. Max Error | $2.326 \pm 0.345$ | $2.144 \pm 0.038$ | $2.077 \pm 0.024$ | $2.083 \pm 0.020$ |
| Vel. IRMSE | $0.212 \pm 0.009$ | $0.238 \pm 0.002$ | $0.237 \pm 0.001$ | $0.234 \pm 0.001$ |
| Vel. BRMSE | $0.520 \pm 0.008$ | $0.564 \pm 0.002$ | $0.561 \pm 0.002$ | $0.564 \pm 0.001$ |
| Vel. Amplitude Ratio | $1.212 \pm 0.017$ | $1.030 \pm 0.004$ | $0.857 \pm 0.003$ | $0.794 \pm 0.002$ |
| Vel. Divergence RMSE (excl. intf.) | $0.387 \pm 0.064$ | $0.374 \pm 0.009$ | $0.375 \pm 0.005$ | $0.370 \pm 0.005$ |
| Vorticity RMSE (excl. intf.) | $1.085 \pm 0.039$ | $1.106 \pm 0.005$ | $1.110 \pm 0.004$ | $1.110 \pm 0.004$ |
| Temp. Relative L2 | $0.031 \pm 0.001$ | $0.032 \pm 0.000$ | $0.034 \pm 0.000$ | $0.035 \pm 0.000$ |
| Temp. Max Error | $14.491 \pm 3.093$ | $16.146 \pm 0.345$ | $16.626 \pm 0.317$ | $16.751 \pm 0.177$ |
| Temp. IRMSE | $2.191 \pm 0.091$ | $2.406 \pm 0.016$ | $2.437 \pm 0.012$ | $2.485 \pm 0.010$ |
| Temp. BRMSE | $1.532 \pm 0.060$ | $1.611 \pm 0.012$ | $1.707 \pm 0.008$ | $1.753 \pm 0.007$ |
| Temp. HF Energy Ratio | $0.931 \pm 0.038$ | $0.897 \pm 0.006$ | $0.931 \pm 0.003$ | $0.944 \pm 0.004$ |
| Wall Heat Flux Rel. Error (%) | $1.522 \pm 0.743$ | $1.930 \pm 0.214$ | $2.243 \pm 0.125$ | $2.452 \pm 0.132$ |

### F.4. Noise on Observations during Inference

We run a noise robustness experiment adding Gaussian noise at four levels to the input observations at inference (Figure 11). Noise levels are: Clean ($\sigma = 0$), Low ($\sigma_{\text{SDF}} = 0.5$, $\sigma_{\text{vel}} = 0.25$), Medium ($\sigma_{\text{SDF}} = 1.0$, $\sigma_{\text{vel}} = 0.5$), and High ($\sigma_{\text{SDF}} = 2.0$, $\sigma_{\text{vel}} = 1.0$). The Low setting is representative of realistic optical flow estimation error and segmentation uncertainty. Results show graceful degradation: velocity and divergence degrade more steeply only at the highest noise level, which substantially exceeds typical optical flow error.

### F.5. Flow Boiling Results

While the main experiments target subcooled *pool boiling*, we additionally evaluate HB-ARFM on a substantially different type of boiling to probe generalization. Table 11 reports the reconstruction results for *flow boiling* over 50 rollout frames sampled at random. Table 12 reports the different rollout steps over 5 random seeds.

## G. Boiling Physics and Governing Equations

Two-phase boiling presents a particularly challenging test case for inverse reconstruction due to the strong coupling between multiple physical processes, sharp discontinuities at moving interfaces, and chaotic bubble dynamics.

*Table 11.* Results for temperature and velocity reconstruction (task 2) in flow boiling using HB-ARFM over 50 random snapshots

| Category | Metric | HB-ARFM |
|---|---|---|
| Velocity Metrics | Vel. Relative L2 | 0.067 |
| | Vel. Max Rel L2 | 0.077 |
| | Vel. Max Error | 6.195 |
| | Vel. IRMSE | 0.160 |
| | Vel. BRMSE | 0.214 |
| | Vel. Amplitude Ratio | 1.004 |
| | Vel. Divergence RMSE (excl. interface) | 0.527 |
| | Vorticity RMSE (excl. interface) | 1.027 |
| Temperature Metrics | Temp. Relative L2 | 1.854 |
| | Temp. Max Rel L2 | 3.285 |
| | Temp. Max Error | 6.107 |
| | Temp. IRMSE | 0.389 |
| | Temp. BRMSE | 0.354 |
| | Temp. HF Energy Ratio | 0.492 |
| | Wall Heat Flux Rel. Error (%) | 0.214 |

*Table 12.* Results for temperature and velocity reconstruction (task 2) in flow boiling across different rollout steps.

| Metric | 1 step | 25 steps | 50 steps | 75 steps | 100 steps |
|---|---|---|---|---|---|
| Vel. Relative $\ell_2$ | $0.0476 \pm 0.0027$ | $0.1145 \pm 0.0016$ | $0.1450 \pm 0.0021$ | $0.1530 \pm 0.0014$ | $0.1531 \pm 0.0010$ |
| Vel. Max Error | $1.7344 \pm 0.1561$ | $2.5005 \pm 0.0505$ | $2.6700 \pm 0.0275$ | $2.7129 \pm 0.0236$ | $2.7057 \pm 0.0164$ |
| Vel. IRMSE | $0.1610 \pm 0.0051$ | $0.3173 \pm 0.0060$ | $0.3867 \pm 0.0060$ | $0.4078 \pm 0.0048$ | $0.4001 \pm 0.0044$ |
| Vel. BRMSE | $0.1453 \pm 0.0106$ | $0.3851 \pm 0.0059$ | $0.4931 \pm 0.0076$ | $0.5128 \pm 0.0049$ | $0.5073 \pm 0.0035$ |
| Vel. Amplitude Ratio | $0.9852 \pm 0.0048$ | $0.9836 \pm 0.0028$ | $1.0025 \pm 0.0021$ | $1.0149 \pm 0.0015$ | $1.0240 \pm 0.0013$ |
| Vel. Divergence RMSE (excl. intf.) | $0.5486 \pm 0.2206$ | $0.5413 \pm 0.0365$ | $0.5493 \pm 0.0163$ | $0.5466 \pm 0.0076$ | $0.5450 \pm 0.0112$ |
| Vorticity RMSE (excl. intf.) | $0.7884 \pm 0.1176$ | $0.8738 \pm 0.0174$ | $1.0060 \pm 0.0126$ | $1.0897 \pm 0.0075$ | $1.1219 \pm 0.0067$ |
| Temp. Relative $\ell_2$ | $3.6334 \pm 0.8289$ | $2.6624 \pm 0.1382$ | $2.1281 \pm 0.0906$ | $2.1188 \pm 0.0514$ | $2.1910 \pm 0.0507$ |
| Temp. Max Error | $2.0071 \pm 0.1463$ | $2.4763 \pm 0.0535$ | $2.7807 \pm 0.0661$ | $2.8806 \pm 0.0288$ | $2.8916 \pm 0.0533$ |
| Temp. IRMSE | $0.3775 \pm 0.0553$ | $0.3684 \pm 0.0122$ | $0.3530 \pm 0.0085$ | $0.3518 \pm 0.0056$ | $0.3549 \pm 0.0052$ |
| Temp. BRMSE | $0.3233 \pm 0.0751$ | $0.3048 \pm 0.0169$ | $0.2771 \pm 0.0122$ | $0.2835 \pm 0.0064$ | $0.2942 \pm 0.0068$ |
| Temp. HF Energy Ratio | $0.2190 \pm 0.0775$ | $0.3489 \pm 0.0405$ | $0.4753 \pm 0.0241$ | $0.4655 \pm 0.0150$ | $0.4418 \pm 0.0132$ |
| Wall Heat Flux Rel. Error (%) | $0.2736 \pm 0.1469$ | $0.1576 \pm 0.0159$ | $0.2250 \pm 0.0159$ | $0.2093 \pm 0.0097$ | $0.1847 \pm 0.0085$ |

## G.1. Physical Phenomena

Boiling involves liquid-to-vapor phase change at a heated surface, forming bubbles that nucleate, grow, merge, and detach unpredictably. The liquid-vapor interface $\Gamma(t)$ is represented by a level-set function $\phi(x, t)$ such that $\Gamma = \{x : \phi(x, t) = 0\}$, with $\phi < 0$ in liquid and $\phi > 0$ in vapor. The interface motion is governed by the balance of buoyancy, surface tension, and phase-change-driven mass flux, creating inherently chaotic multiscale dynamics.

Bubble growth and detachment induce strong flow disturbances in the surrounding liquid, generating turbulent fluctuations and vortex structures in the wake behind rising bubbles. This bubble-induced turbulence significantly enhances mixing and heat transfer compared to single-phase convection.

## G.2. Governing Equations

The system is described by incompressible Navier-Stokes equations coupled with energy transport, solved separately in each phase with jump conditions at the interface.

**Momentum transport (velocity u, pressure $P$):**

$$\frac{\partial \mathbf{u}}{\partial t} + (\mathbf{u} \cdot \nabla)\mathbf{u} = -\frac{1}{\rho'}\nabla P + \frac{1}{\mathrm{Re}}\nabla \cdot \left(\frac{\mu'}{\rho'}\nabla \mathbf{u}\right) + \frac{\mathbf{g}}{\mathrm{Fr}^2} \tag{20}$$

**Energy transport (temperature $\tau$):**

$$\frac{\partial \tau}{\partial t} + \mathbf{u} \cdot \nabla \tau = \frac{1}{\rho' C_p' \mathrm{Pe}} \nabla \cdot (k' \nabla \tau) \tag{21}$$

**Mass conservation with phase change:**

$$\nabla \cdot \mathbf{u} = -\dot{m}\, \mathbf{n}_\Gamma \cdot \nabla \left( \frac{1}{\rho'} \right) \bigg|_\Gamma \tag{22}$$

where $\mathbf{n}_\Gamma = \nabla \phi / |\nabla \phi|$ is the interface surface normal and $\dot{m}$ is the evaporative mass flux. The velocity field is divergence-free within each phase, i.e., $\nabla \cdot \mathbf{u} = 0$, with the exception of the phase interface, where phase change gives rise to a localized source term. The relative thermophysical properties—thermal conductivity $k'$, specific heat capacity $C_p'$, density $\rho'$, and viscosity $\mu'$—are phase-dependent and are defined as ratios between the vapor and liquid property values. The governing equations are expressed in non-dimensional form using the Reynolds Re, Froude Fr, Péclet Pe, and Weber We numbers. The vector $\mathbf{g}$ represents gravitational acceleration. See Hassan et al. (2023); Dhruv et al. (2019) for complete details.

### G.3. Interfacial Coupling

The interface evolves via convection with the interface velocity $\mathbf{u}_\Gamma = \mathbf{u} + (\dot{m}/\rho')\mathbf{n}_\Gamma$:

$$\frac{\partial \phi}{\partial t} + \mathbf{u}_\Gamma \cdot \nabla \phi = 0 \tag{23}$$

The mass flux $\dot{m}$ couples velocity and temperature through the interfacial energy balance:

$$\dot{m} = \frac{\mathrm{St}}{\mathrm{Pe}} \left( k_l' \mathbf{n}_\Gamma \cdot \nabla \tau_l \Big|_\Gamma - k_v' \mathbf{n}_\Gamma \cdot \nabla \tau_v \Big|_\Gamma \right) \tag{24}$$

where $\mathrm{St} = C_{pl} \Delta \tau / Q_l$ is the Stefan number and $Q_l$ is the latent heat of vaporization. This creates tight bidirectional coupling: temperature gradients drive phase change, which generates mass flux at the interface.

### G.4. Challenges for Inverse Reconstruction

Several properties make boiling particularly difficult for machine learning:

**Sharp discontinuities.** Velocity, pressure, and material properties exhibit jump discontinuities at the moving interface, violating smoothness assumptions in many neural architectures.

**Multi-physics coupling.** Temperature and velocity are tightly coupled through the mass flux $\dot{m}$: thermal gradients determine phase change, which creates velocity jumps, which advect temperature. This coupling cannot be decomposed into independent scalar and vector field problems.

**Bubble-induced turbulence.** Rising bubbles generate complex vortex structures and turbulent fluctuations in their wakes. The resulting velocity field exhibits high-frequency temporal variations and fine-scale spatial structures that are difficult to capture from sparse observations.

**Chaotic dynamics.** Bubble nucleation is highly sensitive to thermal fluctuations and surface conditions. Small perturbations in temperature lead to divergent trajectories in interface topology.

**Multiscale structure.** Thermal boundary layers near walls span micrometers while bulk convection occurs over millimeters. Interface thickness is sub-grid-scale while bubble dynamics span the domain.

**Partial observability.** In experiments, only the interface geometry $\phi$ and motion $\mathbf{u}_\Gamma$ are observable via imaging. Temperature and bulk velocity fields are hidden, yet they determine the heat transfer performance (wall heat flux) that engineers care about.

These challenges distinguish boiling from simpler PDE systems (e.g., single-phase flow, pure advection-diffusion) where fields are smooth and physics are weakly coupled. The combination of discontinuities, strong coupling, and turbulence makes boiling an especially stringent benchmark for spatiotemporal inverse modeling.

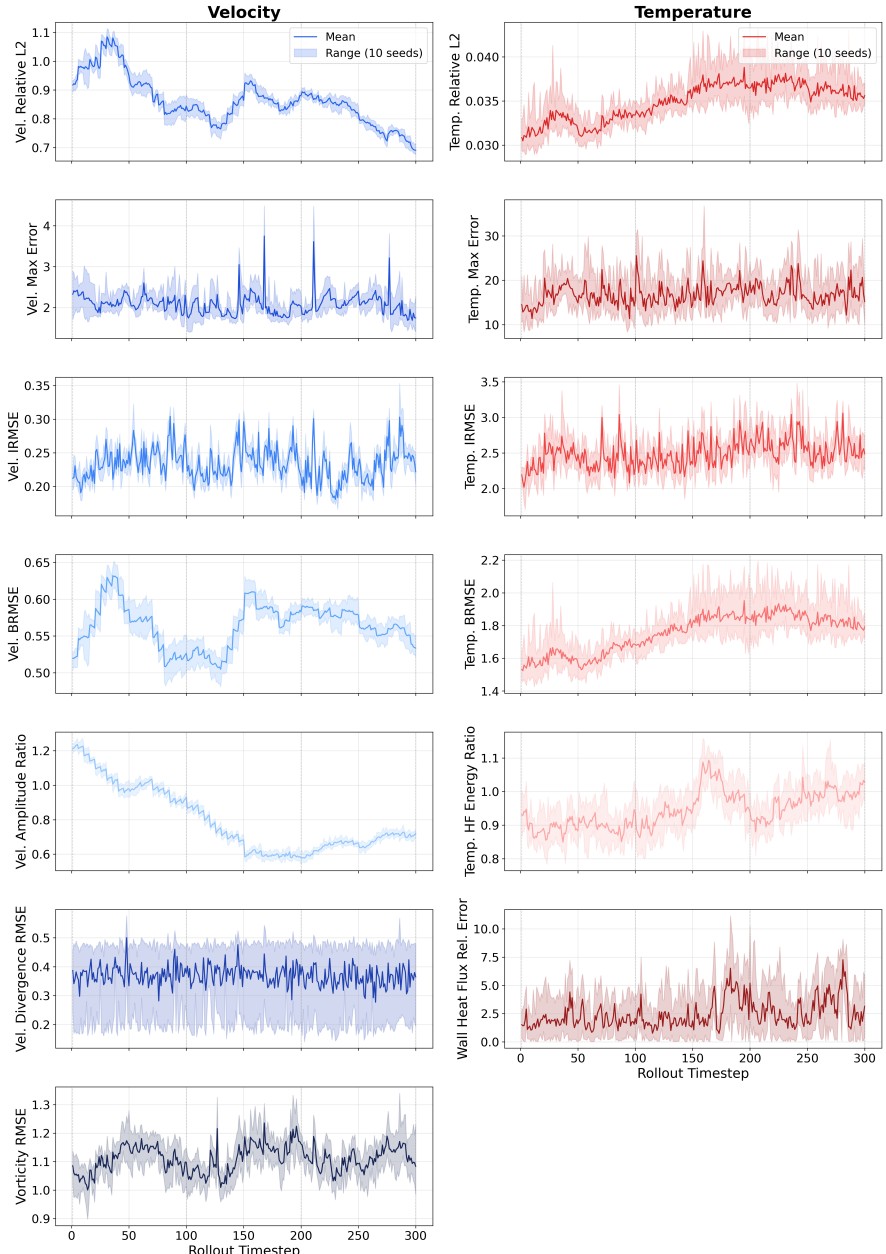

*Figure 9.* **Rollout Metrics for Subcooled Pool Boiling over 300 timesteps.**

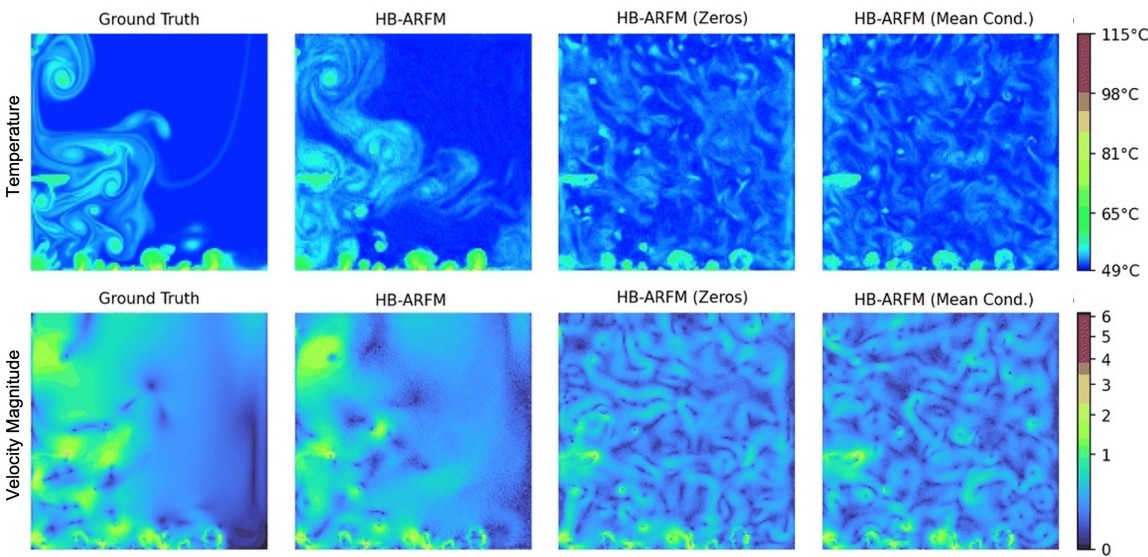

*Figure 10.* **Ablation for Different Initial States.** *HB-ARFM (Zeros)*: Feeds a completely blank initial state. *HB-ARFM (Mean Cond.)*: Averages the observation history over 10 steps.

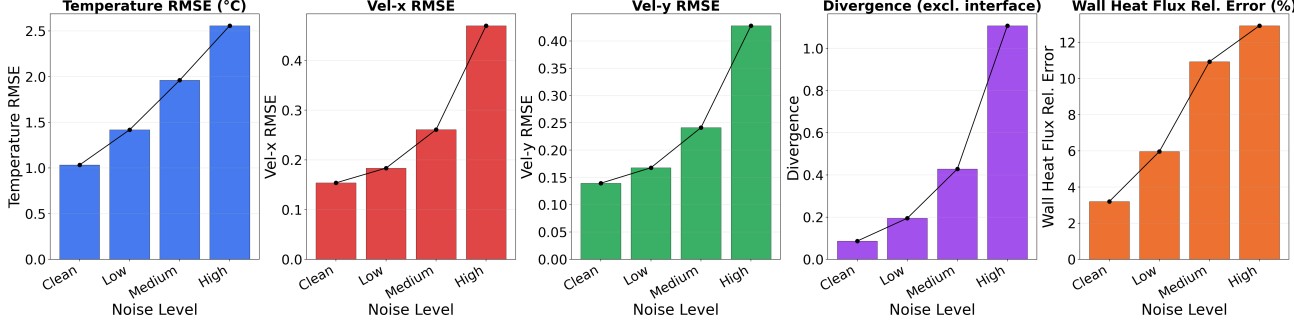

*Figure 11.* **Adding Noise on Observations during Inference.**

