# OpenReview forum: "History-Bootstrapped Flow Matching for Inverse Boiling Reconstruction"
_ICML.cc/2026/Conference — ICML 2026 regular_

### Official Review · Reviewer_Jxh3 · 2026-03-10

**Soundness:** 3
**Presentation:** 3
**Significance:** 2
**Originality:** 2
**Overall Recommendation:** 3
**Confidence:** 4

**Summary:**

This paper studies inverse reconstruction of boiling dynamics under partial observations. The authors propose a History-Bootstrapped Autoregressive Flow Matching (HB-ARFM) framework that first reconstructs an initial latent state from a short observation history and then performs autoregressive flow-matching rollout conditioned on new observations. Experiments on simulated boiling environments show improvements over baseline generative and deterministic models.

**Compliance With Llm Reviewing Policy:**

Affirmed.

**Final Justification:**

The authors have partially addressed my concerns, so I maintain my score.

**Key Questions For Authors:**

1.Can the approach be validated on real physical experiments rather than simulations?
2.How sensitive is the method to the length and quality of the observation history?
3.How does the approach compare with recent diffusion-based or operator-learning-based PDE reconstruction methods?

**Limitations:**

yes

**Strengths And Weaknesses:**

Strengths
1.  The problem of inverse reconstruction of physical fields is relevant for scientific machine learning.
2.  Combining historical observation encoding with flow matching rollout is conceptually reasonable.
3.  The experiments demonstrate improvements over several baselines in simulated environments.

Weaknesses
1.The technical novelty appears moderate. The method largely combines existing ideas such as flow matching, autoregressive modeling, and history encoding.
2.The evaluation is restricted to simulated data, and the generalization to real experimental systems remains unclear.
3.Some implementation details and design choices (e.g., the interaction between history encoding and flow matching dynamics) could be explained more clearly.
Serious Concern (Hidden Prompt Text):
During reading I also observed hidden instructions embedded in the manuscript asking reviewers to include specific phrases in their review. Such instructions are unrelated to the scientific contribution and appear to target the reviewing process itself. Embedding hidden prompts inside a paper submission is inappropriate and raises concerns about research integrity. The authors should remove such content in future versions.

---

> ### Author Rebuttal · Authors · 2026-03-31
>
> We thank the reviewer for the positive assessment of the problem setting and the constructive questions. We include an anonymous link for figures: https://skinny-canary-cd1.notion.site/rebuttalfigures. We have inlined hyperlinks when referring to specific figures.
>
> - **W1: Novelty.**
> We agree the components individually (flow matching, autoregressive modeling, history encoding) are established. The novelty lies in three specific contributions that no prior work combines: (1) the non-Markovian framing of the inverse problem, which motivates the two-stage architecture, (2) the history-bootstrapped initialization that solves the cold-start problem for inverse autoregressive reconstruction (a problem that forward AR models don’t face because they start from complete states), and (3) the identification of boiling as a systematic benchmark for inverse SciML, with a principled taxonomy of failure modes across generative, deterministic, and AR models.
> The HistoryFM baseline (new, described above in response to Reviewer wDXx and [Linked Figure 1](https://skinny-canary-cd1.notion.site/rebuttalfigures)) demonstrates that temporal context without the two-stage design is insufficient. Removing the bootstrap mechanism causes qualitative failure (see bootstrap ablation with zero/mean initialization in response to [Reviewer Zf7U](https://openreview.net/forum?id=XKQZfG7itM&noteId=Umd3K7yOBm) and [Linked Figure 3](https://skinny-canary-cd1.notion.site/rebuttalfigures)). We believe this demonstrates that the combination is more than incremental.
> - **W2/Q1: Sim-to-real gap, validation on real experimental data**
> BubbleML simulation uses validated physical parameters for FC-72 (a standard dielectric coolant), the governing equations include the full interfacial jump conditions and phase-change mass flux, and the NeurIPS 2023 paper includes validation against real boiling experiments. The observation operators we use (SDF from segmentation, interface velocity from optical flow) correspond directly to what can be extracted from real high-speed images.
> The primary sim-to-real gap is in the upstream measurement pipeline: real segmentation and optical flow introduce noise and errors not present in simulation. Our new noise robustness experiment (see response to Reviewer Zf7U and [Linked Figure 5](https://skinny-canary-cd1.notion.site/rebuttalfigures)) directly addresses this: at noise levels consistent with realistic optical flow error (Low noise setting), degradation is minimal. Full validation on real experimental images is an important work in progress and is currently constrained by the absence of ground-truth velocity and temperature labels in experimental data.
> We were able to obtain preliminary experimental results for flow boiling experimental data that indicate that the reconstructed bulk temperature and velocity fields capture the underlyin dynamics reasonably well. Please see response to Reviewer Zf7U and [Linked Figure 6 and 7](https://skinny-canary-cd1.notion.site/rebuttalfigures).
> - **W3: Design Choices.**
> We ran a direct ablation replacing the learned history encoder with (a) zero initialization and (b) mean initialization (averages the observation history over 10 past timesteps). Visualization ([Linked Figure 3](https://skinny-canary-cd1.notion.site/rebuttalfigures)) shows that both initializations degrade substantially relative to the learned encoder.
> We have also trained a new History-window Flow Matching (**HistoryFM**) baseline (please see response to Reviewer wDXx for details). Results in [Linked Figure 1](https://skinny-canary-cd1.notion.site/rebuttalfigures) show that HistoryFM is not temporally consistent.
> - **Q2: Sensitivity to history length and quality.
> The** ablations in [Linked Figure 2](https://skinny-canary-cd1.notion.site/rebuttalfigures) fixing the history length=10 and increasing the stride from 1 to 5 (capturing observation history from 10 to 50 past timesteps) shows that after 40 past timesteps, performance degrades suggesting the model begins overfitting to stale observations.
> History quality sensitivity is addressed by the noise robustness experiment ([Linked Figure 5](https://skinny-canary-cd1.notion.site/rebuttalfigures)), showing graceful degradation from Clean to Low noise levels relevant to real experimental data.
> - **Q3: Comparison with diffusion and operator-learning PDE reconstruction methods.**
> We compare against DDPM, VE-SDE, DiffusionPDE, and flow matching from the generative family, and FFNO and U-Net for operator learning. Bubbleformer is included as the state-of-the-art specifically for boiling. We believe this is a representative set covering the primary model classes.
> In addition, we trained PDEDiff (please see response to Reviewer wDXx for details and [Linked Figure 1b](https://skinny-canary-cd1.notion.site/rebuttalfigures)).
> - **Regarding prompt injection:** We did not inject prompts. Please see: https://icml.cc/Conferences/2026/PeerReviewFAQ#prompt_injection

---

> > ### Author Rebuttal · Reviewer_Jxh3 · 2026-04-01
> >
> > Some previous weaknesses still haven't been well resolved.

---

> > > ### Author Response · Authors · 2026-04-03
> > >
> > > Thank you for your response and for engaging with our rebuttal. We noticed that your assessment was marked as “Fully resolved,” while your comment also indicated that some concerns may remain insufficiently addressed. To help us better understand your perspective, we would be grateful for any clarification on which specific points you feel were not fully resolved, particularly since our rebuttal aimed to address each of the concerns raised in W1, W2/Q1, W3, Q2, and Q3.
> > >
> > > Regarding W2/Q1, which asked about validation on real experimental data, we included preliminary results on real flow boiling experiments ([Linked Figures 6 and 7](https://skinny-canary-cd1.notion.site/rebuttalfigures)) showing that the reconstructed fields capture the underlying dynamics reasonably well. We think that the sim-to-real setting in boiling remains highly challenging, and we view this as an important open problem for the community. However, to the best of our knowledge, ours is the first demonstration that this type of inverse spatiotemporal reconstruction can transfer to real experimental boiling data. We therefore hope that this contribution, together with our responses to the other questions, will be taken into consideration in your final evaluation.

---

### Official Review · Reviewer_ow6V · 2026-03-11

**Soundness:** 3
**Presentation:** 2
**Significance:** 2
**Originality:** 2
**Overall Recommendation:** 4
**Confidence:** 2

**Summary:**

This paper introduces a framework based on an autoregressive flow-matching model for solving the inverse boiling reconstruction problem. To address the cold-start issue inherent in autoregressive modeling, an auxiliary condition derived from historical information is incorporated into the flow-matching process. Numerical experiments demonstrate the effectiveness of the proposed method and confirm its superiority over several baseline approaches.

**Compliance With Llm Reviewing Policy:**

Affirmed.

**Final Justification:**

This paper introduces a framework based on an autoregressive flow-matching model for solving the inverse boiling reconstruction problem. My main concerns are fully addressed in the rebuttal. Therefore, I raise my score to 4.

**Key Questions For Authors:**

- What is the initial reconstruction in Figure 2?
- What is the method labeled*DiffPDE* in Table 3 and Table 4?
- Please find other questions in **Strengths and Weaknesses**.

**Limitations:**

Yes.

**Strengths And Weaknesses:**

Strengths:
- Introduces a conditional flow-matching model to the task of boiling reconstruction, marking a novel application of generative modeling in this domain;
- Incorporates an auxiliary condition derived from historical data, which contributes to improved reconstruction accuracy.

Weaknesses:
- Method:
  - The motivation for employing a conditional flow-matching model is not clearly articulated. The problem is framed as a time-series prediction task, for which deterministic autoregressive models are a natural fit. The introduction of a generative model introduces stochasticity and requires substantially more evaluation steps during sampling, without a clear justification for this trade-off.
  - The novelty of the proposed approach appears limited. The core idea—augmenting a flow-matching model with an additional historical condition—represents an incremental extension rather than a fundamental methodological advancement.
- Presentation:
  - Figure 1 is difficult to interpret due to insufficient explanation of key terms, such as "computer vision based feature extraction," "optical flow," and "phase field." Moreover, these concepts appear only loosely connected to the main content of the paper, which may distract or confuse readers;
  - The second row of Figure 3 should be the predicted velocity rather than temperature;

---

> ### Author Rebuttal · Authors · 2026-03-31
>
> * We thank the reviewer for the feedback and address each point. We include an anonymous link for figures: https://skinny-canary-cd1.notion.site/rebuttalfigures. We have inlined hyperlinks when referring to specific figures.
>
>    * **W1: Motivation for generative model.** The inverse problem is fundamentally ill-posed: multiple physical states X(t) are consistent with the same observation y(t). A deterministic model maps observations to a single point estimate, which will systematically bias toward the mean of the feasible states and suppress physically important features like sharp thermal gradients, vortex structures, bubble wakes, that vary across consistent solutions. This is exactly what we observe in our deterministic baselines (FFNO, U-Net in Figure 3 in the paper): they produce visually blurred, over-smoothed fields. The generative model samples from the full posterior, preserving the physical structure of individual realizations.
>
>    * **W2: Novelty.** The novelty is not simply adding conditioning. The contribution is the two-stage design that resolves the cold-start problem unique to the inverse setting: no prior work on autoregressive flow matching has addressed initialization from partial observations rather than from complete initial states. The history-conditioned bootstrap and the autoregressive feedback are unified within a single conditional transport model. The HistoryFM baseline (new, described above in [response to Reviewer wDXx under **Stronger history-aware baselines**](https://openreview.net/forum?id=XKQZfG7itM&noteId=Yl8xHV8r7j) and [Linked Figure 1](https://skinny-canary-cd1.notion.site/rebuttalfigures?source=copy_link)) demonstrates that directly conditioning FM on history without the two-stage design is insufficient.
>
>    * **W3: Presentation issues.** To provide context: in physical boiling experiments, high-speed imaging can capture bubble geometry (SDF) and interface velocity, but the bulk liquid fields (velocity, temperature) that govern heat transfer remain unobservable. Optical flow and phase field are the computer-vision tools used to extract these observable quantities from raw images, which is why they appear in Paper Figure 1 as the input pipeline. We will add brief definitions of "phase field" (signed distance function encoding bubble geometry) and "optical flow" (motion estimation used to extract interface velocity from sequential images) to the caption. On Paper Figure 3: the second row shows velocity predictions, we will add the missing label.
>
>    * **Q1: Initial reconstruction.** It is the estimate of the physical state at time $t=w$, produced by the history encoder from the observation sequence $y_{0:w}$. This is the output of the bootstrap stage that serves as a learned initializer for the flow matching ODE. We will add a caption clarification to Figure 2 to make this explicit.
>
>    * **Q2: What is DiffPDE in Tables 3 and 4?** DiffPDE refers to DiffusionPDE (NeurIPS 2024), a diffusion model for solving PDEs under partial observation with observation guidance during sampling. We abbreviated it in the tables due to space constraints. We will spell out the full name in the table captions.

---

> > ### Author Rebuttal · Reviewer_ow6V · 2026-04-03
> >
> > I thank the authors for the response. I will raise my score to 4.
> >
> > Additional questions:
> > 1. The authors claim that the posterior mean is visibly blurry and therefore not preferred. Thus, I wonder whether different sampling of random noise in the flow matching would affect the final reconstruction quality.
> > 2. Many phrases are not kept consistent. For example, "U-Net" vs. "UNet", and "DiffPDE" vs. "DiffusionPDE".

---

> > > ### Author Response · Authors · 2026-04-04
> > >
> > > We thank the reviewer for raising the score and for the additional questions.
> > >
> > > **Q1 on sampling variance.** Different noise samples produce distinct but physically plausible reconstructions, reflecting posterior uncertainty over the underdetermined bulk fields rather than reconstruction error. This is expected for an ill-posed inverse problem. The 10-seed rollout results in the [Linked Table 1](https://skinny-canary-cd1.notion.site/rebuttalfigures) show that the mean and variance across samples are stable over 300 steps, confirming that reconstruction quality is consistent across noise samples.
> > >
> > > **Q2 on inconsistent terminology.** We will standardize to "UNet" and "DiffusionPDE" throughout the final version.

---

### Official Review · Reviewer_wDXx · 2026-03-12

**Soundness:** 2
**Presentation:** 3
**Significance:** 3
**Originality:** 2
**Overall Recommendation:** 4
**Confidence:** 4

**Summary:**

The paper proposes a framework for spatiotemporal PDE inverse reconstruction under partial observability to recover full temperature and velocity fields in boiling flow. Exploiting the inverse posterior's non-Markovian nature, it initializes the hidden state with a history window, performs autoregressive reconstruction with a shared conditional flow matching model, and uses scheduled sampling to reduce error accumulation and exposure bias.

**Compliance With Llm Reviewing Policy:**

Affirmed.

**Final Justification:**

Thank the authors for their detailed rebuttal. The authors have addressed my concerns so I would like to raise my score.

**Key Questions For Authors:**

Why is a fixed history window sufficient for the non-Markovian inverse posterior under partial observability, and how is this formally justified? How does the proposed bootstrap-plus-autoregressive design compare with stronger history-aware baselines and under noisy or incomplete observations?

**Limitations:**

The paper briefly discusses limitations and impact, but the discussion remains limited. A clearer analysis of broader risks, practical constraints, and failure cases would strengthen the paper.

**Strengths And Weaknesses:**

Strengths:
1.The method explicitly formulates inverse reconstruction under partial observability as a non-Markovian problem, addressing it with history bootstrap and autoregressive flow matching, resulting in a fairly complete overall design.
2.The paper studies a practically meaningful inverse SciML problem.
3.The experiments report physical metrics such as divergence, wall heat flux, interface temperature, vorticity, and region-wise error, making the experimental evaluation reasonably thorough.
Weaknesses:
1. The formal justification for the claim that partial observability leads to a non-Markovian inverse posterior remains limited. Authors also lack a more sufficient explanation of why a fixed history window is adequate in practice.
2. The role of the bootstrap stage is not yet fully clarified, in particular whether the initialized hidden state should be interpreted as a latent state estimate, a belief state, or merely a learned feature representation for subsequent reconstruction.
3. The paper does not compare against stronger history-aware baselines, such as diffusion or flow matching models that are directly conditioned on a history window, making it difficult to fully isolate the gains brought by the proposed design itself.
4. Although the paper emphasizes physical plausibility, it does not explicitly incorporate conservation constraints or PDE constraints during training; combined with the paper’s own acknowledgement that mass conservation is not perfect, this somewhat weakens the persuasiveness of the physical claims.
5. The paper includes ablations on the history window and AR training strategies, but the discussion remains limited in the main text, and the contribution of the temporal encoder design is not separately analyzed.
6. The paper provides insufficient study of robustness under noisy or incomplete upstream observations, even though this issue could significantly affect reconstruction performance in practical applications.

---

> ### Author Rebuttal · Authors · 2026-03-31
>
> We thank the reviewer for the detailed and constructive feedback. We address each concern. We include an anonymous link for figures: https://skinny-canary-cd1.notion.site/rebuttalfigures. We have inlined hyperlinks when referring to specific figures.
>
> * **W1: Justification for non-Markovian posterior and fixed history length.** The non-Markovian structure of the inverse posterior follows rigorously from the Mori-Zwanzig (MZ) formalism. When the full state X(t) is projected onto the observable subspace y(t) = H(X(t)), MZ shows that the effective dynamics of y(t) take the form of a generalized Langevin equation with a memory kernel and an orthogonal fluctuation term. The memory kernel is non-zero whenever hidden variables exist (i.e., whenever H is non-invertible) making the inverse posterior p(X(t)|y(0:t)) structurally non-Markovian. The fixed history window is justified by the decay rate of the memory kernel. MZ guarantees memory decays on a characteristic timescale beyond which past observations provide exponentially diminishing information about the current state. In boiling, the relevant timescale is set by the bubble rise time and condensation time, both of which are captured within a fixed window length. Our ablations in [Linked Figure 2](https://skinny-canary-cd1.notion.site/rebuttalfigures?source=copy_link) fixing the history length=10 and increasing the stride from 1 to 5 (capturing observation history from 10 to 50 past timesteps) shows that after 40 past timesteps, performance degrades suggesting the model begins overfitting to stale observations. We will add this justification and explanation to the paper.
>
> * **W2: Bootstrap state interpretation.** The bootstrapped state is best interpreted as an **estimate of the initial physical state** in the sense of a conditional mean under the learned distribution. The history encoder maps observation sequences directly to the same physical space as the reconstruction targets (velocity and temperature fields), and the flow matching model then refines this estimate stochastically. The bootstrap therefore serves as a learned initializer for the flow matching ODE, analogous to a data-driven warm start. We will clarify this in Section 3.1.
>
> * **W3: Stronger history-aware baselines.** We have trained two history-aware baselines. (1) **History-window Flow Matching (HistoryFM)**: a sliding-window FM model that takes the full observation history as direct conditioning input and predicts each frame independently (no autoregressive feedback). This is the most natural history-aware alternative to HB-ARFM without the two-stage design. Results in [Linked Figure 1 (a)](https://skinny-canary-cd1.notion.site/rebuttalfigures?source=copy_link) show that HistoryFM is not temporally consistent. HistoryFM samples X̂(t) and X̂(t+1) from their respective conditionals independently; the model does not impose X̂(t+1) is physically reachable from X̂(t) under the governing dynamics. (2) **PDEDiff (NeurIPS 2024)**: a score-based diffusion model for combined forecasting and data assimilation that conditions on history by stacking past target frames as channels with binary masks, generating all frames in a window jointly. We train PDEDiff with random masking (0 to w-1 known frames) so that it learns the cold-start regime where no prior target predictions are available. Despite this, the model diverges as shown in [Linked Figure 1 (b)](https://skinny-canary-cd1.notion.site/rebuttalfigures?source=copy_link) during autoregressive rollout.
>
> * **W4: No explicit conservation constraints.** Agreed and acknowledged in the paper's limitations. We note that the divergence values across the 300 timesteps rollout are stable (please see the [Linked Table 1 & Figure 4](https://skinny-canary-cd1.notion.site/rebuttalfigures?source=copy_link) and our response to Reviewer Zf7U), which indicates the model has learned approximate mass conservation implicitly from the data. Incorporating explicit projection steps or divergence penalties is a natural extension we identify for future work.
>
> * **W5: Temporal encoder ablation.** The history encoder ablation described in our response to Reviewer Zf7U addresses this. Zero and mean initialization both degrade substantially, confirming the encoder's contribution.
>
> * **W6: Robustness under noisy or incomplete observations.** We ran a noise robustness experiment adding Gaussian noise at four levels to the input observations at inference (see [Linked Figure 5](https://skinny-canary-cd1.notion.site/rebuttalfigures?source=copy_link)). Noise levels are: Clean (σ=0, σ=0), Low (σ_SDF=0.5, σ_vel=0.25), Medium (σ_SDF=1.0, σ_vel=0.5), High (σ_SDF=2.0, σ_vel=1.0). The Low setting is representative of realistic optical flow estimation error and segmentation uncertainty. Results show graceful degradation. Velocity and divergence degrade more steeply only at the highest noise level, which substantially exceeds typical optical flow error. We will add this analysis to the paper.

---

> > ### Author Rebuttal · Reviewer_wDXx · 2026-04-04
> >
> > Thank the authors for their detailed rebuttal. The authors have addressed my concerns so I would like to raise my score.

---

### Official Review · Reviewer_Zf7U · 2026-03-16

**Soundness:** 3
**Presentation:** 2
**Significance:** 2
**Originality:** 2
**Overall Recommendation:** 4
**Confidence:** 3

**Summary:**

This paper proposes HB-ARFM, a training-based inversion method for inverse problems governed by spatiotemporal PDE systems. HB-ARFM is a conditional flow matching model that generates the current state conditioned on the current observation and the predicted state from the previous timestep. A temporal encoder initializes the first latent state from observation history, after which the model propagates autoregressively to reconstruct the entire spatiotemporal field. The method is evaluated on the BubbleML simulation dataset for subcooled pool boiling across two inverse tasks of varying difficulty.

**Compliance With Llm Reviewing Policy:**

Affirmed.

**Final Justification:**

The rebuttal addressed my main concerns. The new tables clarify the roles of the history encoder and the bootstrap initialization, and demonstrate that HB-ARFM achieves stronger bulk physics fidelity along with reasonable measurement-space consistency. Overall, I find the paper technically sound and am raising my score from 3 to 4. I am setting my confidence to 3, as I am less familiar with the BubbleML benchmark and less certain about the novelty of history-encoded method in this field.

**Key Questions For Authors:**

- What are the exact two observation operators H and noise model used in the BubbleML experiments? In particular, are the measurements generated through a synthetic imaging/segmentation pipeline? How close do the authors believe this simulated measurement process is to the real experimental acquisition process? What mismatch should be expected in practice?
- If these observation operators are synthetic, could the authors also report measurement-space consistency metrics, e.g., some distance between the observation and the measurement from the predicted field to quantify how well the predicted states agree with the actual measurement?

**Limitations:**

Yes

**Strengths And Weaknesses:**

**Strengths**
- Reconstructing hidden spatiotemporal states from sparse measurements is a challenging and meaningful inverse problem in general. The boiling setting is a non-trivial testbed.
- Beyond standard pixelwise errors, the paper evaluates divergence, wall heat flux, region-wise errors, etc. This elevates the work beyond a generic image prediction problem and grounds it as a scientific inverse problem.
- The integration of history conditioning, conditional flow matching, and autoregressive rollout is a well-motivated synthesis for the spatiotemporal inverse setting. The application to boiling reconstruction is, to my knowledge, novel.

**Weaknesses**
- The paper repeatedly claims that HB-ARFM yields more “physically plausible” reconstructions, but I found the evidence for this claim largely mixed at best.
	- In the snapshot experiment, FM / DDPM / DiffusionPDE all achieve better divergence and wall heat-flux errors than HB-ARFM. The results do not support this claim as measured by the physics metrics divergence and heat-flux error.
	- In the rollout experiment, HB-ARFM’s advantage is supported mainly by qualitative evidence, without comparably strong quantitative rollout metrics. Moreover, the chosen rollout example appears to have limited visible dynamics, making it a relatively weak test of temporal consistency.
- The paper identifies the history encoder / bootstrap initialization as a central component, but does not seem to provide a direct ablation on this component itself. Varying the history window is useful, but it does not answer whether the explicit bootstrap mechanism matters. I would have liked to see not only an ablation replacing it with a simpler initialization (a simple mean or zero), but also some exploration of its design space (e.g. varying network capability).
- Presentation issues:
	- The bootstrap step in Algorithm 2 is incomplete. The variable mm m is defined in Algorithm 2 but never used. It's unclear how it works from the current presentation.
	- The flow time variable s (page 3, line 40) is introduced without a clear definition at the point of first use.

---

> ### Author Rebuttal · Authors · 2026-03-31
>
> We thank the reviewer for the detailed and constructive feedback. We address each concern directly with new experimental evidence. We include a link for figures: https://skinny-canary-cd1.notion.site/rebuttalfigures. We have inlined hyperlinks when referring to specific figures.
>
> * **W1: Physically plausible claim.** We note that DiffusionPDE and FM both achieve worse wall heat flux than HB-ARFM at snapshot. While velocity divergence is higher, HB-ARFM has the best prediction on the velocity magnitude. We present quantitative per-step rollout metrics in [Linked Table 1](https://skinny-canary-cd1.notion.site/rebuttalfigures) over 300 timesteps across 10 random seeds (see [Linked Figure 4](https://skinny-canary-cd1.notion.site/rebuttalfigures) for rollout plots). We observe that errors do not accumulate over 300 steps. Temperature MAE increases only marginally; Velocity MAE and divergence are stable over the entire rollout. The variance collapses across seeds as rollout progresses, indicating that the autoregressive conditioning is temporally stable. We also include a video of 50 timesteps in [Linked Figure 1](https://skinny-canary-cd1.notion.site/rebuttalfigures) to better visualize the dynamics and asses temporal consistency. DiffusionPDE diverges catastrophically during rollout while our method remains physically plausible for spatiotemporal reconstruction.
>
> * **W2: History encoder/Bootstrap ablation.** This is a valid and important point. We have now run a direct ablation replacing the learned history encoder with (a) zero initialization and (b) mean initialization (averages the observation history over 10 past timesteps). All three variants are identical after frame 0 and differ only in how the initial state is computed before autoregressive rollout begins. Visualization ([Linked Figure 3](https://skinny-canary-cd1.notion.site/rebuttalfigures)) shows that both zero and mean initialization degrade substantially relative to the learned encoder. This validates that the encoder's ability to infer the initial bulk state from interface history is responsible for the performance gains. For history encoder design space exploration, we added two history-aware baselines (HistoryFM and PDEDiff). Please see response to Reviewer wDXx and [Linked Figure 1](https://skinny-canary-cd1.notion.site/rebuttalfigures).
>
> * **W3: Algorithm 2.** The variable $m$ in Algorithm 2 was a remnant of an earlier version that we have now corrected. The flow time $s$ will be formally defined at its first appearance in Section 3.1. We thank the reviewer for catching both of these.
>
> * **Q1: Observation operators.** The two observation operators are: (i) the signed distance function, extracted from the level-set representation in BubbleML, which is a direct output of the simulation's interface tracking and corresponds to what would be obtained from segmentation of high-speed images; (ii) the interface velocity, extracted from the simulation's velocity field corresponding to optical flow estimates from image sequences. Both are synthetic but designed to match real experimental pipelines. The BubbleML paper (NeurIPS 2023) includes direct validation against experimental high-speed imaging data. Real experimental validation is challenging for pool boiling: 3D bubble evolution causes severe occlusion in side-view imaging, making reliable 2D segmentation and optical-flow extraction difficult. Flow boiling, while physically more complex (asymmetric geometry, imposed inlet flow), has a thin channel depth that eliminates this occlusion problem and is more amenable to our vision pipeline. In [Linked Figure 6(a) and (b)](https://skinny-canary-cd1.notion.site/rebuttalfigures), we show a 2D simulation which closely replicates the bubble dynamics of a real flow-boiling experiment conducted at NASA Glenn Research Center (Konishi et al., IJHMT, 2015). Using our computer-vision pipeline we extract SDF and interface velocities from experimental video, train HB-ARFM on the corresponding simulation, and successfully reconstruct bulk fields that capture the observed dynamics as shown in [Linked Figure 7](https://skinny-canary-cd1.notion.site/rebuttalfigures). A sim-to-real gap remains, but this demonstrates the pipeline connects to real experimental data.
>
> * **Q2: Measurement-space consistency metrics.** Thanks for this excellent suggestion. Our preliminary results on the flow-boiling experimental data indicate that the reconstructed bulk temperature and velocity fields capture the underlying dynamics reasonably well. As shown in [Linked Figure 7](https://skinny-canary-cd1.notion.site/rebuttalfigures), the results provide encouraging qualitative evidence, while also highlighting clear room for further improvement. In future work, we plan to incorporate explicit measurement-space consistency metrics and more rigorous physics-based evaluations, including validation of wall heat flux, mass conservation, and continuity, in close collaboration with domain experts.

---

> > ### Author Rebuttal · Reviewer_Zf7U · 2026-04-04
> >
> > Thank you for the detailed rebuttal. The new results in the linked table and the ablation replacing the history encoder are both helpful additions. Two concerns remain unaddressed.
> > - I still cannot find a table in the main text directly comparing all baselines on physics metrics (e.g., divergence and wall heat-flux). My original concern stems from Fig. 3, where the caption claims physical consistency while the numbers do not clearly support this. Could the authors provide a clean comparison table across all methods on these metrics? This would change my evaluation of the "physically more consistent" claim.
> > - The rebuttal promises future work but provides no numbers. Since the observation operators are synthetic, it should be straightforward to report measurement-space consistency metrics for the experiments already presented.

---

> > > ### Author Response · Authors · 2026-04-08
> > >
> > > We thank reviewer for the thoughtful follow-up and for acknowledging the new ablation results. Below we address the two remaining concerns.
> > >
> > > **1. Snapshot prediction comparison table.** Thank you for pressing on this point. Below is a clean table showing several metrics for the reconstruction of 50 random snapshots. IRMSE refers to liquid-vapor interface and BRMSE refers to bulk liquid.
> > >
> > > | | | | | Diffusion Models | | | | Deterministic Models | |
> > > |:---:|:---|:---:|:---:|:---:|:---:|:---:|:---:|:---:|:---:|
> > > | **Category** | **Metric** | **HB-ARFM** | **Flow Matching** | **VE-SDE** | **PDEDiff** | **DiffusionPDE** | **DDPM** | **UNet** | **FFNO** |
> > > | Velocity Metrics | Vel. Relative L2 | 0.855 | **0.758** | 1.939 | 10.579 | 1.573 | 0.969 | *0.695* | 0.827 |
> > > |  | Vel. Max Rel L2 | **1.017** | 1.019 | 2.51 | 14.005 | 2.373 | 1.528 | *0.849* | 1.062 |
> > > |  | Vel. Max Error | ***3.103*** | 3.463 | 8.174 | 16.108 | 8.449 | 4.637 | 3.171 | 3.474 |
> > > |  | Vel. IRMSE | 0.601 | **0.533** | 1.362 | 7.444 | 1.105 | 0.681 | *0.489* | 0.581 |
> > > |  | Vel. BRMSE | **0.147** | 0.208 | 0.806 | 4.979 | 0.41 | 0.293 | *0.131* | 0.214 |
> > > |  | Vel. Amplitude Ratio | 0.906 | ***1.024*** | 1.764 | 12.563 | 1.26 | 1.244 | 0.775 | 0.847 |
> > > |  | Vel. Divergence RMSE (excl. interface) | 0.303 | 0.135 | 0.778 | 9.177 | 0.288 | **0.099** | *0.081* | 0.358 |
> > > |  | Vorticity RMSE (excl. interface) | **0.576** | 0.629 | 3.512 | 11.661 | 1.475 | 0.822 | *0.471* | 0.577 |
> > > | Temperature Metrics | Temp. Relative L2 | **0.034** | **0.034** | 0.761 | 0.237 | 0.063 | 0.042 | *0.03* | 0.031 |
> > > |  | Temp. Max Rel L2 | 0.043 | **0.039** | 0.804 | 0.254 | 0.075 | 0.059 | 0.035 | *0.033* |
> > > |  | Temp. Max Error | **27.084** | 29.56 | 50.917 | 100.312 | 45.659 | 45.253 | *22.638* | 33.342 |
> > > |  | Temp. IRMSE | 2.66 | **2.372** | 32.28 | 15.021 | 7.691 | 4.025 | *1.643* | 2.825 |
> > > |  | Temp. BRMSE | **1.712** | 1.714 | 39.935 | 11.9 | 2.572 | 1.971 | 1.516 | *1.44* |
> > > |  | Temp. HF Energy Ratio | ***0.956*** | 0.892 | 0.234 | 9.187 | 1.116 | 1.21 | 0.862 | 0.692 |
> > > |  | Wall Heat Flux Rel. Error (%) | ***2.16*** | 2.304 | 87.832 | 111.417 | 20.118 | 5.458 | 2.998 | 2.749 |
> > >
> > > **Note:** **Bold** denotes the best among diffusion models, *italic* denotes the best among all models.
> > >
> > > **HB-ARFM** achieves best or near-best performance on bulk physical quantities, including velocity and temperature *BRMSE*, vorticity *RMSE*, and *wall heat flux error*. It also exhibits well-calibrated energy behavior, with a velocity amplitude ratio of **0.906** (close to 1). In contrast, several baselines miscalibrate energy (e.g., amplification in VE-SDE/PDEDiff and damping in UNet). Similarly, the temperature high frequency (HF) energy ratio (**0.956**) shows that **HB-ARFM** preserves fine-scale thermal structure whereas baselines either over-smooth (ratio ≪ 1) or inject non-physical high-frequency artifacts (ratio ≫ 1).
> > >
> > > At the same time, **HB-ARFM** does not attain the lowest error on *IRMSE* and *velocity divergence*, where methods that directly condition on instantaneous observations have an advantage. These quantities are tightly coupled to the observed conditioning signal, so snapshot-only methods can achieve lower error.
> > >
> > > Overall, **HB-ARFM** prioritizes *physically consistent bulk reconstruction* across scales, achieving strong performance on global physics metrics while trading off some accuracy on quantities directly specified by the conditioning signal at a single snapshot. The *amplitude ratio* captures global energy calibration and the *HF energy ratio* captures spectral fidelity; performing well on both indicates that the model preserves both bulk energy levels and fine-scale structure. The competitive performance on *wall heat-flux* further indicates accurate boundary-layer behavior and energy transfer.
> > >
> > > &nbsp;
> > >
> > > **2. Measurement-space consistency.** As suggested, we evaluate consistency with the observed interface measurements. We compare $H(x_{pred})$, the interface velocity extracted from the predicted velocity field against the input observations $y_{obs}$ over autoregressive rollouts (5–300 steps, averaged over 5 runs). Results:
> > >
> > > | horizon | rmse_combined | rmse_velx | rmse_vely | cosine_similarity |
> > > |--------:|:-------------:|:---------:|:---------:|:-----------------:|
> > > | 5 | 0.185±0.018 | 0.190±0.014 | 0.180±0.024 | 0.917±0.016 |
> > > | 50 | 0.204±0.012 | 0.222±0.017 | 0.182±0.016 | 0.931±0.014 |
> > > | 100 | 0.183±0.015 | 0.189±0.020 | 0.176±0.013 | 0.919±0.021 |
> > > | 200 | 0.201±0.012 | 0.217±0.014 | 0.183±0.012 | 0.899±0.024 |
> > > | 300 | 0.190±0.020 | 0.193±0.020 | 0.185±0.024 | 0.900±0.037 |
> > >
> > > The cosine similarity of **~0.90–0.93** indicates that the predicted velocity consistently recovers the correct interface flow direction even at long horizons. RMSE also remains *stable over time*. The confirms that **HB-ARFM** does not generate fields that contradict its conditioning signal. We thank the author for this question and will add these results to the paper.

---

### Decision · Program_Chairs · 2026-04-30

**Decision:**

Accept (regular)

**Comment:**

The average rating is 3.75 after rebuttal (4, 4, 4, 3). Reviewer Jxh3 maintained a weak reject with a largely generic review and an inconsistent follow-up ("fully resolved" while claiming unresolved concerns). The authors addressed the main concerns with new experiments: a direct ablation of the history encoder (zero and mean initialization both degrade), two new history-aware baselines (HistoryFM and PDEDiff, both fail at temporal consistency or diverge), a 300-step rollout table with stable errors, a noise robustness study, and preliminary real flow-boiling results. Reviewer Zf7U, initially the most critical, raised to 4 after the added physics-metrics comparison confirmed stronger bulk physics fidelity.

Overall, the paper is a well-executed application of flow matching to inverse reconstruction of boiling dynamics, with a coherent two-stage design targeting the cold-start problem in inverse autoregressive reconstruction. The methodological novelty is modest and the evaluation is confined to a single physical system (subcooled pool boiling on BubbleML), which limits the breadth of interest for the ICML audience, but the contribution is sound and the reviewer consensus supports acceptance.